# Yeast homologs of human MCUR1 regulate mitochondrial proline metabolism

Mohammad Zulkifli [1], John K. Neff[1], Shrishiv A. Timbalia [1], Natalie M. Garza[1], Yingqi Chen[2], Jeramie D. Watrous[2], Marta Murgia [3,4], Prachi P. Trivedi[1], Steven K. Anderson[1], Dhanendra Tomar [5,6], Roland Nilsson [7,8,9], Muniswamy Madesh[10], Mohit Jain[2] & Vishal M. Gohil[1✉]

Mitochondria house evolutionarily conserved pathways of carbon and nitrogen metabolism that drive cellular energy production. Mitochondrial bioenergetics is regulated by calcium uptake through the mitochondrial calcium uniporter (MCU), a multi-protein complex whose assembly in the inner mitochondrial membrane is facilitated by the scaffold factor MCUR1. Intriguingly, many fungi that lack MCU contain MCUR1 homologs, suggesting alternate functions. Herein, we characterize *Saccharomyces cerevisiae* homologs Put6 and Put7 of MCUR1 as regulators of mitochondrial proline metabolism. Put6 and Put7 are tethered to the inner mitochondrial membrane in a large hetero-oligomeric complex, whose abundance is regulated by proline. Loss of this complex perturbs mitochondrial proline homeostasis and cellular redox balance. Yeast cells lacking either Put6 or Put7 exhibit a pronounced defect in proline utilization, which can be corrected by the heterologous expression of human MCUR1. Our work uncovers an unexpected role of MCUR1 homologs in mitochondrial proline metabolism.

[1] Department of Biochemistry and Biophysics, Texas A&M University, College Station, TX 77843, USA. [2] Departments of Medicine and Pharmacology, University of California, San Diego, 9500 Gilman Avenue, La Jolla, CA 92093, USA. [3] Department of Biomedical Sciences, University of Padova, 35121 Padua, Italy. [4] Max-Planck-Institute of Biochemistry, Martinsried 82152, Germany. [5] Department of Medical Genetics and Molecular Biochemistry, Lewis Katz School of Medicine at Temple University, Philadelphia, PA 19140, USA. [6] Center for Translational Medicine, Lewis Katz School of Medicine at Temple University, Philadelphia, PA 19140, USA. [7] Cardiovascular Medicine Unit, Department of Medicine, Karolinska Institutet, SE-171 76 Stockholm, Sweden. [8] Division of Cardiovascular Medicine, Karolinska University Hospital, SE-171 76 Stockholm, Sweden. [9] Center for Molecular Medicine, Karolinska Institutet, SE-171 76 Stockholm, Sweden. [10] Department of Medicine, Cardiology Division, Center for Precision Medicine, University of Texas Health Science Center at San Antonio, San Antonio, TX 78229, USA. ✉email: vgohil@tamu.edu

Mitochondria are ancient organelles that house many evolutionarily conserved biochemical pathways critical to energy generation, intermediary metabolism, calcium homeostasis, stress response, and apoptosis[1,2]. Accordingly, aberrant mitochondrial function has been associated with numerous human diseases such as cancer, diabetes, neurodegenerative disorders, and monogenic metabolic disorders[3,4]. The importance of mitochondria in fundamental cellular functions and human health has motivated systematic studies to define the mitochondrial proteome[5–8]. Despite these advances, functional annotation of many of the evolutionarily conserved proteins lags behind. This limits our understanding of basic mitochondrial biology and the molecular basis of many human diseases emanating from mitochondrial dysfunction[9].

To address this challenge, we have focused on deciphering the functions of uncharacterized mitochondrial proteins that are evolutionarily conserved and thus most likely to be involved in the fundamental mitochondrial processes of energy generation and intermediary metabolism. Here, we investigated the role of two uncharacterized but highly conserved mitochondrial proteins from *Saccharomyces cerevisiae*, Fmp32 and Ylr283w, which we have renamed Put6 and Put7, respectively. Both of these proteins share significant sequence homology to the mammalian mitochondrial calcium uniporter regulator 1 (MCUR1), which has been shown to positively regulate mitochondrial calcium ($Ca^{2+}$) uptake[10]. MCUR1 has been shown to physically interact with the mitochondrial $Ca^{2+}$ uniporter (MCU), where it functions as a scaffold factor for hetero-oligomeric MCU complex assembly[10,11]. These studies suggested a role of MCUR1 in regulating mitochondrial bioenergetics via MCU-mediated $Ca^{2+}$ signaling. However, phylogenomic analysis of MCU and its regulatory partners showed that some yeast species, including *S. cerevisiae*, have lost MCU during evolution[12], suggesting that yeast homologs of MCUR1 function in processes other than mitochondrial $Ca^{2+}$ signaling.

Here, we use a combination of cell biological and biochemical methods, as well as nutrient-sensitized metabolomics to systematically analyze the function of Put6 and Put7 in mitochondrial intermediary metabolism. We demonstrate that Put6 and Put7 are orthologues of human MCUR1 and are mitochondrial matrix-facing integral membrane proteins that form a large hetero-oligomeric complex required for proline utilization.

## Results

**Put6 and Put7 share homology to human MCUR1.** To identify the *S. cerevisiae* homologs of human MCUR1 (UniProt id: Q96AQ8-1), we performed a domain enhanced lookup time accelerated BLAST (DELTA-BLAST) analysis. This identified two yeast proteins belonging to the DUF1640 family, Fmp32 (Put6) and Ylr283w (Put7), with a query coverage of 54–55% and identity ranging from 18 to 25% (Supplementary Fig. 1a). A reciprocal DELTA-BLAST analysis using Put6 and Put7 proteins as query sequences yielded MCUR1 as the best hit, suggesting a high degree of sequence conservation (Supplementary Fig. 1b). To survey the distribution of MCUR1 across different phyla in eukaryotes, we performed phylogenetic analysis of the MCUR1 protein in 15 model organisms. This showed that, unlike MCU, MCUR1 homologs are present in many fungi, including *S. cerevisiae* (Fig. 1a). Multiple sequence alignment of Put6, Put7, MCUR1, and its paralog CCDC90B showed that the majority of conserved residues are in the predicted C-terminal DUF1640 region (Supplementary Fig. 1c). A secondary structure prediction analysis of Put6 and Put7 identified a mitochondrial targeting sequence, a coiled-coil domain, and a transmembrane domain similar to their human counterparts (Fig. 1b).

**Put6 and Put7 are inner mitochondrial membrane proteins.** Previous high throughput protein localization studies in yeast identified Put6 and Put7 in mitochondria[7,13,14]. Using differential centrifugation, we confirmed that Put6 and Put7 localize predominantly to mitochondria by comparing the enrichment profiles of the C-terminal V5-tagged versions of these proteins to that of porin (Por1), a known mitochondrial protein (Fig. 1c, d). We recovered Put6 and Put7 in the membrane-containing fraction of an alkaline carbonate extraction, along with the integral inner mitochondrial membrane protein Cox2 but unlike Coa6, the soluble protein of the mitochondrial intermembrane space (Fig. 1e, f). Next, we performed a protease-protection assay to investigate the membrane topology of the Put6 and Put7 proteins. Put6 and Put7 were accessible to proteinase K only after solubilization of mitochondrial membranes with Triton X-100, suggesting that the soluble domain(s) of these proteins are facing the mitochondrial matrix in a manner similar to the mitochondrial matrix-facing Tim44 (Fig. 1g, h). The differences we observed in proteinase K sensitivity between Put6 and Put7 during swelling buffer treatment may be due to the longer C-terminal tail of Put7 (Fig. 1b), which may expose the V5 tag to the IMS and promote proteinase K degradation. Taken together, these results suggest that Put6 and Put7 are integral membrane proteins of the inner mitochondrial membrane with the N-terminal soluble domain(s) facing the matrix (Fig. 1i).

**Put6 and Put7 form a large hetero-oligomeric complex.** The CCDC domain is known to mediate protein-protein interactions through coiling of helices within or between polypeptide chains[15]. To test whether Put6 and Put7 form a higher-order complex in mitochondrial membranes, we performed Blue-native polyacrylamide gel electrophoresis (BN-PAGE) analysis of mitochondria isolated from V5-tagged Put6 or Put7 expressing cells. We found that both Put6 and Put7 migrated as a part of a very large (~720 kDa) protein complex (Fig. 2a), and loss of either of these proteins prevented complex formation (Fig. 2b). Consistent with a requirement for both proteins in the formation of the high molecular weight complex, we observed that the deletion of Put7 led to a complete loss of Put6 protein (Supplementary Fig. 2a). Similarly, steady-state levels of Put7 were markedly reduced in *put6Δ* cells (Supplementary Fig. 2b). Since the molecular weight of the Put6/Put7-containing complex is similar to the mitochondrial respiratory chain (MRC) supercomplexes, we wondered whether Put6 and Put7 associate with MRC supercomplexes. The assembly and abundance of Put6-containing complex were not impacted by the loss of MRC supercomplexes in *bcs1Δ* and *shy1Δ* yeast mutants, which lack the assembly factor for MRC complexes III and IV, respectively (Supplementary Fig. 2c).

The molecular weight of the Put6/Put7-containing complex (~720 kDa) was several-fold higher than the combined molecular weight of Put6 and Put7 (62 kDa). We, therefore, considered the possibility that the higher-order complex contains other protein subunits. To define the components of this higher-order complex, we performed immunoprecipitation (IP) of the complex, followed by quantitative mass spectrometry (MS). MS analysis of three independent isolates of Put6-V5 immunoprecipitates identified Put6 and Put7 as the two most enriched proteins (Fig. 2c). When using Put7-V5 as the bait protein, we again identified Put6 and Put7 as the most enriched proteins (Fig. 2d). Notably, no additional proteins common to both IP experiments were identified in a statistically significant manner in the conditions tested, suggesting that the high-order complex is most likely a hetero-oligomeric complex of Put6 and Put7.

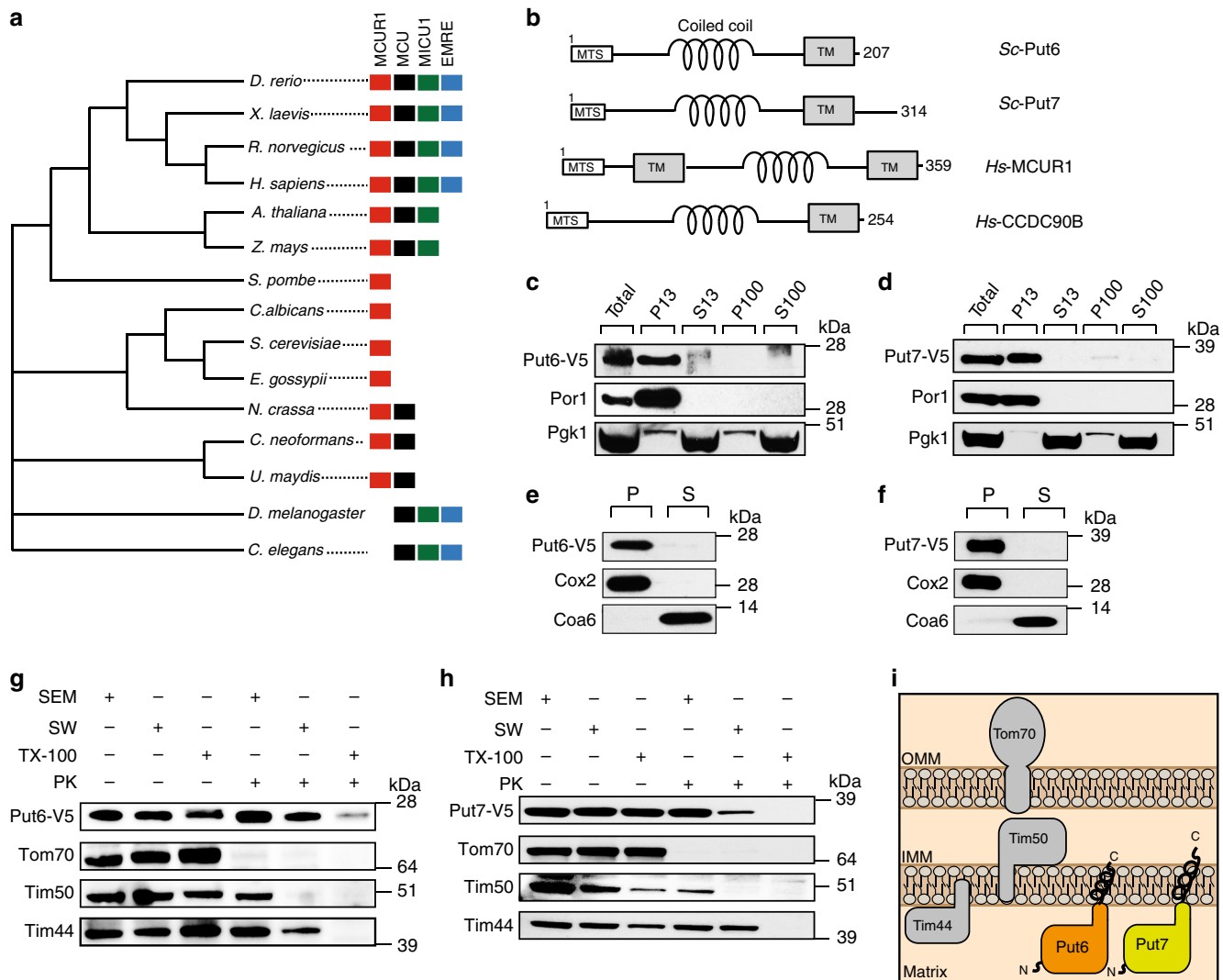

**Fig. 1 Phylogeny, localization and topology of Put6 and Put7. a** Phylogenetic tree focused on 15 selected species across different eukaryotic phyla showing the presence of MCUR1 and mitochondrial Ca²⁺ uniporter components MCU, MICU1, and EMRE. **b** Schematic showing predicted domain organization of Put6, Put7, MCUR1, and CCDC90B proteins. MTS, mitochondria targeting sequence; TM, transmembrane helix; *Sc*, *Saccharomyces cerevisiae*; *Hs*, *Homo sapiens*. **c**, **d** Put6 and Put7 are localized to the mitochondrial fraction. SDS-PAGE immunoblot analysis of different subcellular fractions prepared from Put6-V5 and Put7-V5 expressing cells. Por1 and Pgk1 were used as mitochondrial and cytosolic markers, respectively. Whole-cell protein lysate is depicted as "Total", P and S refer to pellet and supernatant, respectively, and the number following P and S refers to the centrifugation speed × 1000 *g*. Data are representative of two independent experiments. **e**, **f** Put6 and Put7 are integral membrane proteins. Mitochondria containing Put6-V5 and Put7-V5 were exposed to alkaline treatment at pH 11.5. Membrane integrated (P, pellet) and soluble (S) material were separated by centrifugation and analyzed by immunoblotting. Cox2 and Coa6 were used as markers for integral membrane and soluble proteins, respectively. Data are representative of two independent experiments. **g**, **h** N-terminal soluble domains of Put6 and Put7 are matrix facing. Mitochondria expressing Put6-V5 and Put7-V5 were resuspended in isotonic (SEM), hypotonic-swelling (SW), and Triton X-100 (TX-100)-containing buffer and were treated with (+) or without (−) proteinase K (PK). After 30 min incubation, samples were analyzed by immunoblotting using antibodies against V5 and the marker proteins, Tom70 (OMM), Tim50 (IMS), and Tim44 (matrix). Data are representative of three independent experiments. **i** Model showing the localization and topology of Put6 and Put7 proteins. OMM, outer mitochondrial membrane; IMM, inner mitochondrial membrane. Source data are provided as a Source data file.

To confirm the physical interaction between Put6 and Put7, we performed a co-immunoprecipitation experiment using mitochondria isolated from *put7Δ* cells expressing C-terminal hemagglutinin (HA)-tagged Put7, and from *put6Δput7Δ* yeast cells co-expressing Put6-V5 and Put7-HA from their endogenous promoters. IP with anti-V5 antibody did not precipitate Put7-HA, which demonstrated antibody specificity; however, IP with anti-V5 antibody did recover both Put6-V5 and Put7-HA, corroborating the physical interaction between Put6 and Put7 (Fig. 2e). A similar result was also observed by doing a reciprocal IP experiment using anti-HA antibody where

immunoprecipitation of Put7-HA results in a strong co-immunoprecipitation of Put6-V5 (Fig. 2f).

**Put6 and Put7 are not required for MRC function.** Previous studies have reported conflicting findings regarding the function of Put6 and its human homolog MCUR1 in MRC biogenesis. Paupe et al. showed that MCUR1 is required for MRC complex IV biogenesis in human cell lines and that Put6 is required for respiratory growth of yeast cells[16]. However, Tomar et al. did not observe any perturbation in any MRC complex subunits in

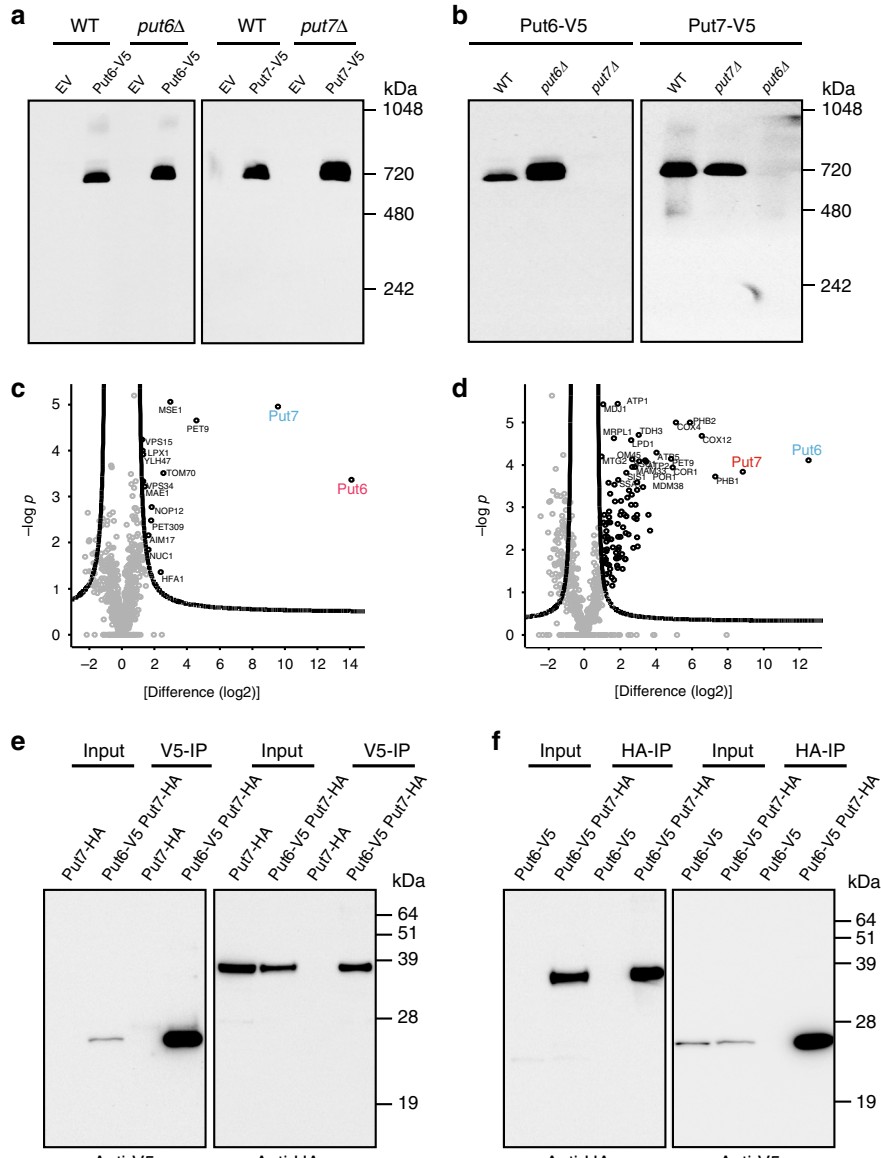

**Fig. 2 Put6 and Put7 form a large hetero-oligomeric complex. a** Put6 and Put7 are part of a large protein complex. Digitonin-solubilized mitochondria from BY4741 WT, *put6Δ*, or *put7Δ* yeast cells transformed with empty vector, Put6-V5 or Put7-V5, respectively, were resolved by Blue-native (BN) PAGE and analyzed by immunoblotting using anti-V5 antibody. Data are representative of three independent experiments. **b** Put6 and Put7 complex formation is dependent on each other. Mitochondria from the indicated yeast strains expressing either Put6-V5 or Put7-V5 were solubilized in a digitonin-containing buffer and analyzed by BN-PAGE/immunoblotting. Data are representative of three independent experiments. **c, d** Volcano plot showing the enrichment of Put6 and Put7 interacting proteins. Immunoprecipitates of digitonin-solubilized mitochondria expressing either Put6-V5 (**c**), or Put7-V5 (**d**) were analyzed by mass spectrometry. The abundance of Put6-V5 and Put7-V5 interacting proteins were normalized using immunoprecipitate obtained from cells transformed with empty vector. **e, f** Put6 and Put7 physically interact with each other. Mitochondria expressing the indicated proteins were solubilized in 1% digitonin-containing buffer and incubated with either anti-V5 (**e**) or anti-HA (**f**) antibodies. Input lysate (1% of mitochondrial extract) and anti-V5/anti-HA immunoprecipitates were analyzed by immunoblotting using anti-HA or anti-V5 antibodies. Source data are provided as a Source data file.

MCUR1 deficient cells[11]. To systematically decipher the role of Put6 and Put7 in MRC function, we performed nutrient-sensitized growth assays by culturing *put6Δ* and *put7Δ* cells on different carbon sources. The growth of *put6Δ* and *put7Δ* mutants in glucose-containing fermentable medium (YPD) or glycerol/ethanol-containing non-fermentable media (YPG, YPE, and YPGE) were indistinguishable from wild-type (WT) yeast cells (Fig. 3a). Because it has previously been reported that *put6Δ* cells exhibit reduced growth on agar plates with a non-fermentable carbon source at elevated temperature[16], we performed growth measurements of WT, *put6Δ*, and *put7Δ* single and double mutants at 30 and 37 °C in three different non-fermentable

carbon sources (YPG, YPE, and YPGE) on agar plates. Again, in these growth conditions, we did not detect any difference in the growth of mutants as compared to WT cells (Fig. 3b). To confirm that this phenotype is not yeast strain-specific, we deleted *PUT6* in three different genetic backgrounds and performed a similar growth-based assay. Under these conditions, the growth of WT and *put6Δ* cells in all three genetic backgrounds was comparable (Supplementary Fig. 3). The respiratory-competent growth of *put6Δ* and *put7Δ* cells suggests that the MRC is functional in these mutants.

Since growth-based assays do not always reveal subtle biochemical defects, we decided to measure the abundance and

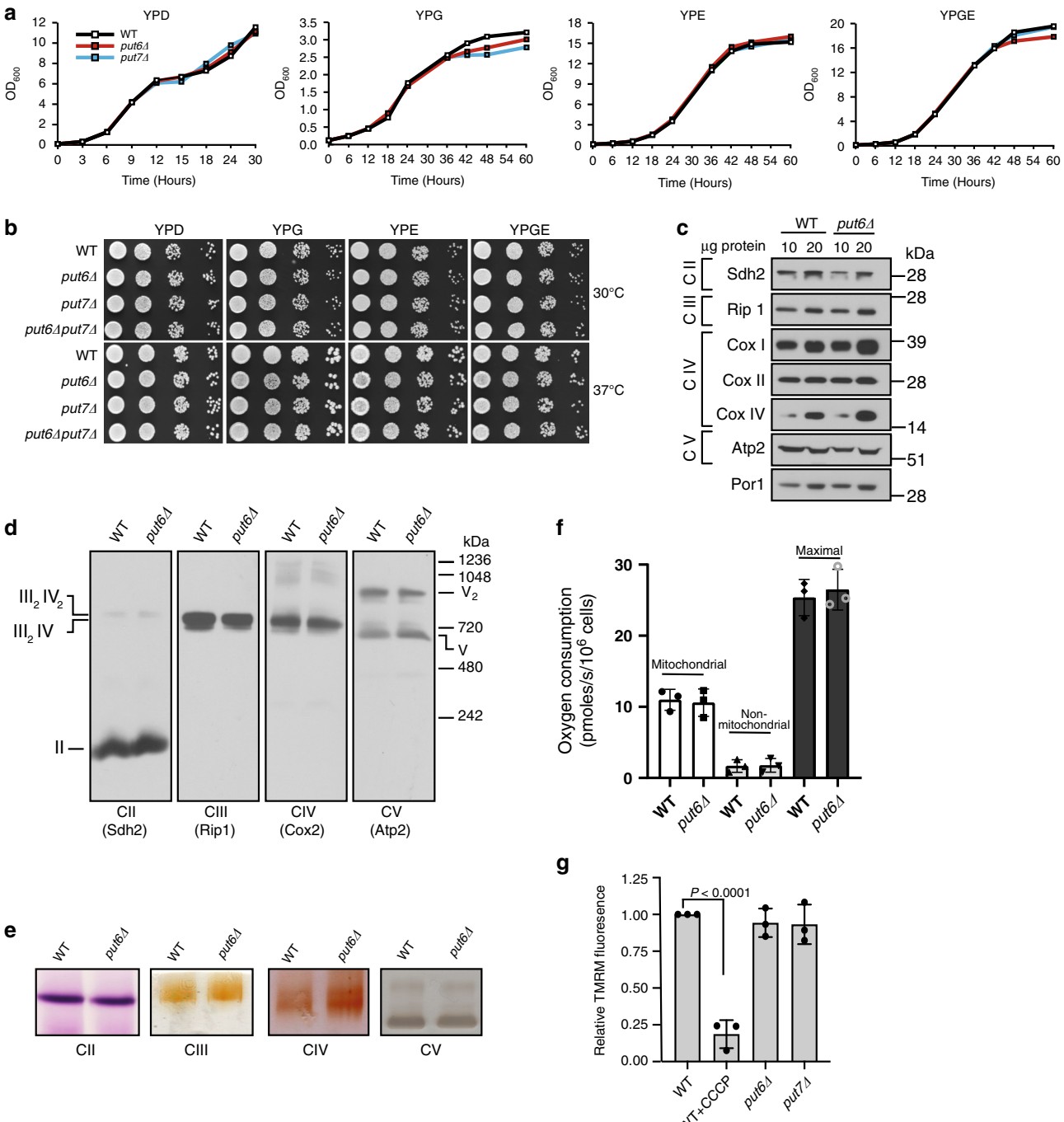

**Fig. 3 Put6 and Put7 are not required for MRC function or formation. a, b** Put6 and Put7 are not required for respiratory growth. **a** Growth analysis of WT, *put6Δ*, and *put7Δ* yeast cells in YPD, YPG, YPE, and YPGE at 30 °C. Absorbance at 600 nm was measured at the indicated time intervals. Data are representative of three independent experiments. **b** Ten-fold serially diluted WT, *put6Δ*, *put7Δ*, and *put6Δput7Δ* yeast cells were seeded on the indicated plates at 30 °C (upper panel) and 37 °C (lower panel). Pictures were taken after 2 (YPD) or 5 (YPG, YPE, and YPGE) days of growth. Data are representative of at least three independent experiments. **c** Mitochondrial proteins were extracted from WT and *put6Δ* yeast cells and analyzed by SDS-PAGE/western blotting using antibodies specific for MRC complexes II–V. Por1 was used as a loading control. Data are representative of three independent experiments. **d** Mitochondria were solubilized in 1% digitonin and analyzed by BN-PAGE immunoblotting. The molecular weights and stoichiometry of the supercomplexes are indicated. Data are representative of three independent experiments. **e** Digitonin-solubilized mitochondrial complexes from WT and *put6Δ* cells were resolved by CN-PAGE, followed by in-gel activity staining for complexes II–V. Data are representative of three independent experiments. **f** Oxygen consumption of YPGE grown WT and *put6Δ* cells was measured using high-resolution O2K Fluorespirometer at 30 °C. Mitochondrial respiration refers to antimycin sensitive respiration, non-mitochondrial respiration refers to antimycin resistant respiration, and maximal respiratory rates were calculated after addition of 5 μM CCCP. **g** Measurement of mitochondrial membrane potential using TMRM fluorescence in WT, *put6Δ*, and *put7Δ* cells grown in YPGE media. Data were normalized to WT control and are expressed as mean ± SD (*n* = 3 biologically independent experiments). Statistical significance was assessed by one-way analysis of variance (ANOVA) with Tukey's multiple comparison test. WT and mutants used in this panel are from BY4741 genetic background (see Supplementary Table 1). Source data are provided as a Source data file.

assembly of MRC complexes in mitochondria isolated from WT and $put6\Delta$ cells by performing SDS-PAGE/BN-PAGE followed by Western blotting. Neither the levels of individual respiratory chain complex subunits nor the formation of respiratory supercomplexes was impaired by the loss of Put6 (Fig. 3c, d). In fact, we observed an elevation in the levels of MRC complex IV subunits (Fig. 3c). To determine if the MRC complexes were functional, we measured their activities by in-gel activity assays and found that the activities were not reduced in $put6\Delta$ cells (Fig. 3e). Consistent with the in-gel activity assays, we did not observe any perturbation in the mitochondrial oxygen consumption under basal or stimulated conditions (Fig. 3f). Mitochondrial membrane potential in the mutants was also comparable to that of WT cells, which implies that respiratory chain is fully functional (Fig. 3g). Taken together, these data clearly show that loss of Put6 or its homolog Put7 does not impair MRC function or formation.

**Put6 and Put7 are required for proline utilization**. In addition to intermediary carbon metabolism, mitochondria also play a critical role in nitrogen metabolism. Therefore, we decided to test whether Put6 and Put7 might be involved in utilizing amino acids as nitrogen sources. It has been previously shown that yeast *S. cerevisiae* can use most amino acids as the sole nitrogen source[17,18]. To test the ability of yeast cells to utilize individual amino acids as nitrogen sources, it is necessary to use prototrophic *S. cerevisiae* strains, because commonly used auxotrophic laboratory strains require nutrient supplements including amino acids that can also be used as nitrogen sources. To this end, we constructed $put6\Delta$ and $put7\Delta$ in prototrophic WT background and grew them in a defined minimal media using galactose as the carbon source and different amino acids as sole nitrogen sources. In these experiments, we chose to use galactose as a carbon source because unlike glucose, it promotes respiro-fermentative growth and does not inhibit mitochondrial biogenesis in *S. cerevisiae*[19,20]. We used ammonium sulfate, a commonly used nitrogen source for laboratory yeast growth, as a control. WT and mutants grew similarly in synthetic media containing ammonium sulfate as the sole nitrogen source (Fig. 4a). Of all the twenty proteinogenic amino acids tested, both $put6\Delta$ and $put7\Delta$ cells showed a growth defect only when provided with proline as the sole nitrogen source, suggesting an impaired proline metabolism in these mutants (Fig. 4a). Consistent with a previous study[21], we found that histidine, lysine, and cysteine cannot be used by yeast as the sole nitrogen sources (Fig. 4a). We also performed growth measurements on agar plates with glucose or galactose as the carbon sources and ammonium sulfate or proline as the sole nitrogen source and found that the single or double deletions of *PUT6* and *PUT7* exhibited similar growth defects in proline-containing media that was independent of the carbon source used (Fig. 4b, c). However, this growth defect was less pronounced than that of $put1\Delta$ cells, which are unable to catalyze the first step in proline catabolism (Fig. 4b, c).

Put1 and Put2, the two enzymes required for proline utilization, are highly upregulated in yeast when proline is used as a sole nitrogen source[22]. To test whether growth on proline could upregulate the levels of Put6 and Put7 in a similar manner, we measured the steady-state levels of functionally active V5-tagged Put6 and Put7 proteins (Supplementary Fig. 4a) from WT cells grown in media containing ammonium sulfate, proline, or glutamate as the sole nitrogen source. We found that the steady-state protein level of Put6 was ~20-fold higher in proline medium compared to media containing ammonium sulfate or glutamate (Fig. 4d). A significant increase in the protein levels of Put7, which appears to be constitutively expressed in all media

conditions, was also observed in proline medium (Fig. 4e). An increase in steady-state protein levels of Put6 and Put7 also correlated with an increase in the formation of the Put6/Put7 complex as demonstrated by BN-PAGE/western blot analysis (Fig. 4f). In addition to the ~720 kDa complex, we also observed at least three additional Put7-containing low molecular weight complexes formed in the presence of proline, which became apparent when the blot was exposed for an extended period (Fig. 4f). These additional lower molecular weight bands may represent the precursor forms of the functional complex. To determine if Put6 and Put7 are regulated at the transcriptional level, we performed quantitative RT-PCR analysis in WT prototrophic yeast grown in the presence of proline or ammonium sulfate as sole nitrogen sources. As reported earlier[23], we observed a striking ~50-fold increase in *PUT1* expression in proline-containing media; however, there was no change in the mRNA levels of *PUT6* and *PUT7* in the presence of proline (Supplementary Fig. 4b), suggesting that these genes are not regulated transcriptionally. These data are consistent with the observation that the promoter regions of *PUT6* and *PUT7* do not contain proline-specific upstream activation sequences for the transcriptional activator Put3, which is known to upregulate genes involved in proline utilization[24]. Collectively, these data demonstrate that Put6 and Put7 are specifically required for growth using proline as a sole nitrogen source and their abundance is regulated by proline at the post-transcriptional level.

**Loss of Put6 and Put7 alters mitochondrial proline uptake**. Proline catabolism has been extensively studied in the yeast *S. cerevisiae*[25]. Proline enters the mitochondria via an unknown transporter, where it is converted to $\Delta^1$-pyrroline-5-carboxylate (P5C) by proline dehydrogenase (Put1) in a FAD/FADH$_2$ coupled reaction (Fig. 5a). The cyclic P5C intermediate either undergoes a spontaneous reaction to form glutamate-γ-semialdehyde (GSA) or is exported to the cytosol as a part of the P5C-proline cycle (Fig. 5a). The proposed role of this cycle is to maintain the balance of redox equivalents between mitochondria and the cytosol by reducing P5C back to proline in a reaction catalyzed by P5C reductase (Pro3), which uses cytosolic NADPH as the electron donor. GSA formed from proline in the mitochondria is converted to glutamate by P5C dehydrogenase (Put2). The amino group of glutamate is then released by NAD$^+$-dependent glutamate dehydrogenase 2 (Gdh2) to serve as a nitrogen source for the biosynthesis of other macromolecules. Alternatively, glutamate may serve as a biosynthetic precursor for arginine (Fig. 5a).

To identify the step(s) in which Put6 and Put7 might be operating, we performed liquid chromatography-mass spectrometry (LC-MS) based metabolic profiling on cellular extracts from WT and $put6\Delta$ cells grown in ammonium sulfate, proline or glutamate as the sole nitrogen source. When cultured in a proline-containing medium, we observed a striking accumulation of proline in $put6\Delta$ cells compared to WT cells (Fig. 5b and Supplementary Fig. 5a). Interestingly, downstream metabolites including glutamate, N-acetyl glutamate, ornithine, citrulline, arginine, and glutamine were also significantly elevated, suggesting that loss of Put6 did not disrupt mitochondrial proline uptake (Fig. 5c–i and Supplementary Fig. 5b–h). To directly measure mitochondrial proline uptake, we performed an in vitro $^{14}$C proline uptake assay in isolated mitochondria. We found a significantly higher rate of proline uptake (Fig. 5j), which is consistent with our in vivo metabolomics data. Next, we wondered whether Put6/Put7 had any role in P5C-proline cycling. Therefore, we measured P5C efflux from isolated

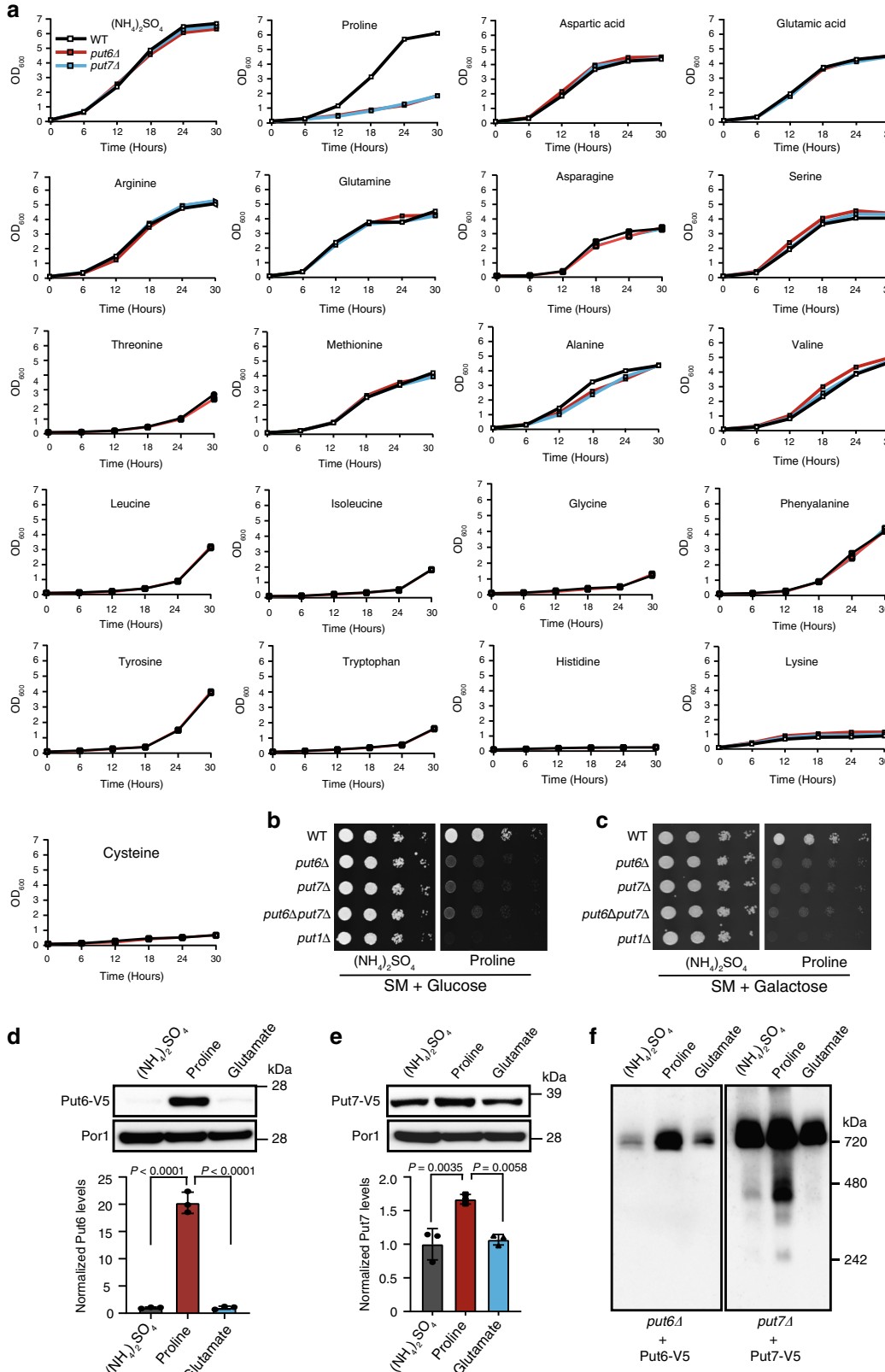

mitochondria supplied with proline and found a higher rate of P5C export, which suggested faster cycling (Fig. 5k). Increased P5C/proline cycling may divert proline away from glutamate synthesis in mitochondria; therefore, we measured the rate of glutamate accumulation as well as glutamate export from mitochondria. Consistent with elevated levels of glutamate and its downstream metabolites seen in our metabolomics data, we observed an increased rate of glutamate accumulation in $put6\Delta$ mitochondria (Fig. 5l) while glutamate export was not perturbed (Fig. 5m). Together, these data suggest that Put6/Put7 act as a negative regulator of mitochondrial proline uptake.

**Fig. 4 Put6 and Put7 are specifically required for growth in proline. a** Prototrophic WT, *put6Δ*, and *put7Δ* yeast cells were cultured at 30 °C in synthetic liquid media containing the indicated amino acids as the sole nitrogen source and galactose as the carbon source. Growth was monitored by measuring absorbance at 600 nm at the indicated time intervals. Ammonium sulfate was used as the most favored nitrogen source. **b, c** Ten-fold serially diluted yeast cells were seeded on synthetic media (SM) plates with glucose (**b**) or galactose (**c**) as carbon source and ammonium sulfate or proline as the sole nitrogen source. Proline oxidase, Put1, which is required for growth in proline-containing media, was used as a control. Pictures were taken after 3 days of seeding. **d, e** Put6 and Put7 protein levels increase in proline-containing media. Mitochondria expressing Put6-V5 and Put7-V5 were isolated from cells grown in indicated nitrogen source and subjected to immunoblot analysis using anti-V5 antibody. The relative abundance of Put6-V5 and Put7-V5 were quantified by densitometric analysis using ImageJ software. Data were normalized to protein levels in ammonium sulfate and expressed as mean ± SD (*n* = 3 biologically independent experiments). Statistical significance was assessed by one-way analysis of variance (ANOVA) with Tukey's multiple comparison test. **f** 20 μg of digitonin-solubilized mitochondrial fractions from (**d, e**) were resolved by BN-PAGE and immunoblotted using anti-V5 antibody. *put7Δ* + Put7-V5 blot was overexposed to show the sub complexes. All the above data are representative of at least three independent measurements. WT and mutants used in this panel are from prototrophic ACY genetic background (see Supplementary Table 1). Source data are provided as a Source data file.

**Loss of Put6 and Put7 perturbs cellular redox balance**. Since proline cycling and catabolism are coupled to cellular redox balance, we decided to interrogate cellular levels of NADH/NAD$^+$ and NADPH/NADP$^+$ and mitochondrial reactive oxygen species (ROS) production. Loss of Put6 and Put7 did not impact the levels of NADH, NAD$^+$, NADPH, and NADP$^+$ in cells cultured in ammonium sulfate (Supplementary Fig. 6a–d), and consequently, there was no alteration in the NADH/NAD$^+$ and NADPH/NADP$^+$ ratios (Fig. 6a, b). However, when *put6Δ* and *put7Δ* mutants were cultured in proline, the levels of both NADH and NADPH increased without any perturbation in the levels of NAD$^+$ and NADP$^+$ (Supplementary Fig. 6e–h). Thus, the NADH/NAD$^+$ and NADPH/NADP$^+$ ratios were elevated in *put6Δ* and *put7Δ* mutants when grown in proline as the sole nitrogen sources (Fig. 6c, d). Concomitantly, we observed increased mitochondrial ROS levels in *put6Δ* and *put7Δ* mutants only in proline-containing medium (Fig. 6e, f). Interestingly, overexpression of Put6 and Put7 also impaired growth in proline and led to an increase in mitochondrial ROS levels, suggesting that optimal levels of these proteins are necessary for maintaining mitochondrial ROS homeostasis (Supplementary Fig. 6i, j). To further corroborate the increased ROS levels detected by MitoSOX, we measured the oxidative inactivation of mitochondrial aconitase and ROS induced mitochondrial fragmentation[26,27]. Aconitase activity in isolated mitochondria was drastically reduced in the mutants when grown in proline-containing medium (Fig. 6g, h). We also observed increased mitochondrial fragmentation in *put6Δ* and *put7Δ* mutants when proline was used as the sole nitrogen source (Fig. 6i). We reasoned that the growth defect of *put6Δ* and *put7Δ* mutants in proline could be due to increased ROS; however, the growth defect was not mitigated with glutathione supplementation, suggesting that ROS is not the primary cause for defective growth in proline-containing media (Fig. 6j).

Interestingly, we found that the growth defect of *put6Δ* and *put7Δ* in proline-containing media could be rescued by the addition of alternate nitrogen sources, including glutamate and arginine (Fig. 6k, l). We used glutamate and arginine because they are favorable nitrogen sources (Fig. 4a) and are downstream products of proline oxidation (Fig. 5a). The growth rescue with glutamate or arginine supplementation suggests that growth retardation of *put6Δ* and *put7Δ* is not due to the toxicity of proline or proline-derived metabolites, and that these mutants retain the ability to resume growth upon the availability of a favorable nitrogen source(s) (Fig. 6k, l). Furthermore, cell-cycle analysis of WT and mutants grown in proline-containing media shows that the majority of *put6Δ* and *put7Δ* cells fail to enter the G2 phase of the cell cycle and remain arrested in the G0/G1 phase, a hallmark of nitrogen limitation (Fig. 6m)[28,29].

**Human MCUR1 functionally complement Put6 and Put7 mutants**. To test the functional similarity between human MCUR1 and yeast homologs, we performed a complementation experiment by heterologous expression of codon-optimized human MCUR1 in *put6Δ*, *put7Δ*, and *put6Δput7Δ* yeast cells, and tested for the ability to rescue the proline-specific growth defects. As expected, constructs expressing the yeast Put6 and Put7 were able to rescue the growth of their respective knockouts (Fig. 7a). Interestingly, human MCUR1 was able to rescue the growth of *put6Δ* and *put7Δ*, as well as *put6Δput7Δ* mutants (Fig. 7a). SDS-PAGE western blot analysis on mitochondria isolated from MCUR1-expressing yeast cells demonstrates that MCUR1 is localized to the yeast mitochondria and also undergoes post-translational processing, as inferred from the ~28 kDa band instead of the expected ~39 kDa band based on its primary structure (Fig. 7b). Notably, similar post-translational modification of native MCUR1 has been observed in mammalian cell lines where the processed form lacked the first 140 amino acid residues at N-terminus[11,30]. To understand the biochemical basis of MCUR1-mediated rescue, we measured proline levels in *put6Δ* and *put7Δ* single and double mutants and found that MCUR1 completely restored WT levels of proline in these mutants (Fig. 7c). These results suggest that MCUR1 has retained the functionality of its ancestral homologs Put6 and Put7 with respect to their role in proline metabolism.

## Discussion

Here we characterized yeast mitochondrial proteins Fmp32 and YLR283w, which we renamed Put6 and Put7 owing to their specific requirement in proline utilization. We show that Put6 and Put7 physically interact to form a large hetero-oligomeric complex in the inner mitochondrial membrane, where their levels are dependent on each other and the presence of proline as a sole nitrogen source. Loss of Put6 and Put7 leads to accumulation of proline, its downstream metabolites, and ultimately growth arrest, a phenotype that can be rescued by expression of their human homolog MCUR1.

We chose to focus on Put6 and Put7 because of their homology to MCUR1, which has been implicated in MCU-dependent mitochondrial Ca$^{2+}$ homeostasis and MRC complex IV biogenesis[10,11,16]. The yeast *S. cerevisiae* genome encodes MRC complex IV, but unlike other eukaryotes, it does not possess MCU[12]. Therefore, we expected an MRC complex IV-specific role of Put6/Put7 in mitochondrial function. However, we did not observe any defect in the activity or the assembly of MRC complexes in yeast cells lacking Put6 (Fig. 3). In fact, we observed elevated levels of MRC complex IV subunits (Fig. 3c), an observation that mirrors findings in human cells lacking MCUR1[31]. In order to rule out a strain- and condition-specific role of Put6 in mitochondrial respiratory functions, we constructed Put6 knockout yeasts in three different genetic backgrounds and tested

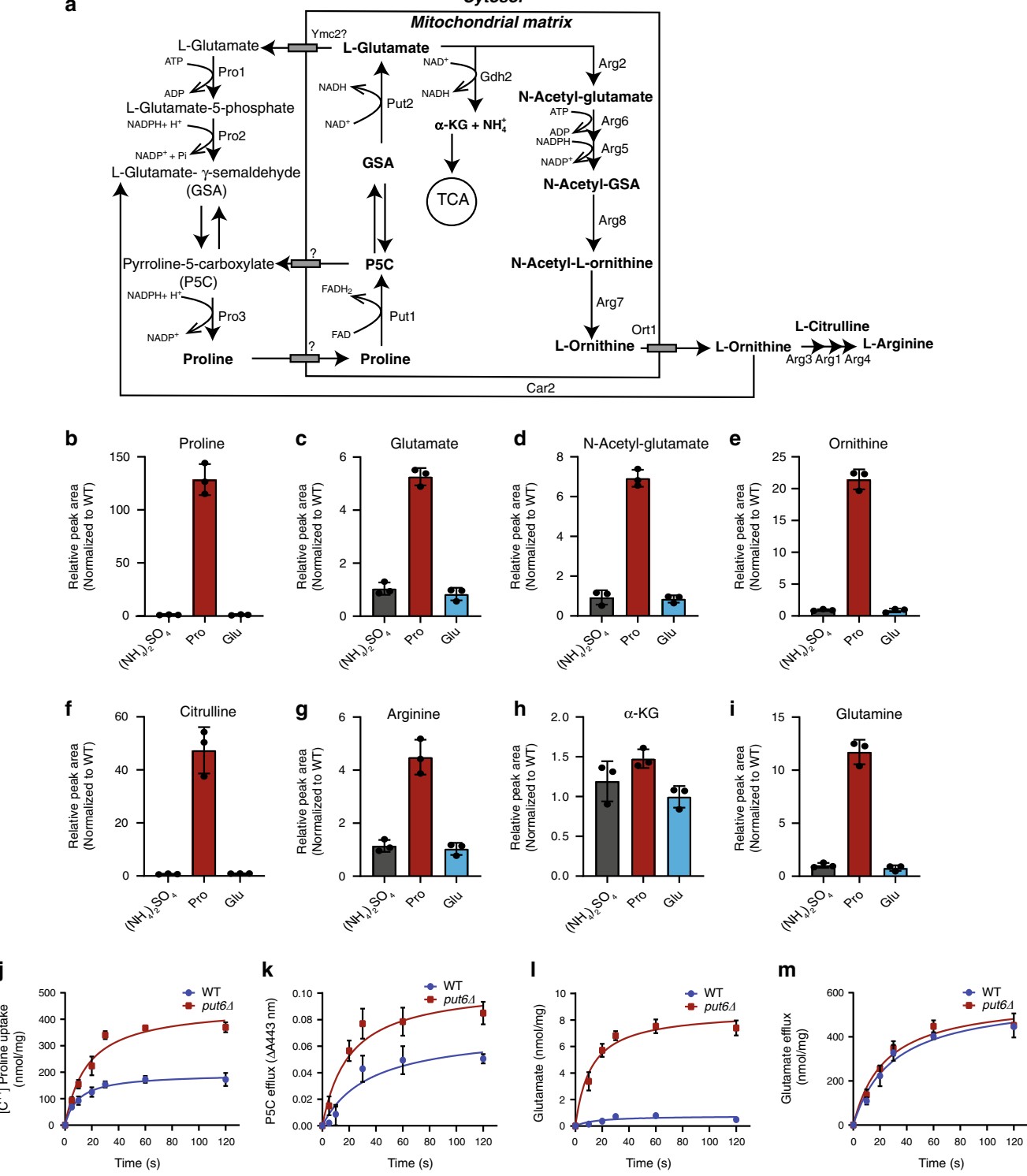

**Fig. 5 Metabolomic analysis shows increased levels of proline in *put6Δ* yeast cells. a** A biochemical pathway depicting proline metabolism in *S. cerevisiae*. Proline catabolism occurs in mitochondria and is highlighted in bold letters. **b–i** LC-MS-based quantification of proline (**b**), glutamate (**c**), N-acetyl glutamate (**d**), ornithine (**e**), citrulline (**f**), arginine (**g**), α-ketoglutarate (**h**), and glutamine (**i**) in *put6Δ* yeast cells cultured in indicated nitrogen source (Pro —proline, Glu—glutamate). Data were normalized to WT and plotted as mean ± SD ($n = 3$ biologically independent experiments). **j** Time course of [14C] proline uptake in isolated mitochondria from ACY WT and *put6Δ* yeast cells grown in proline as the sole nitrogen source. All the data points were normalized to time zero and plotted as mean ± SD ($n = 3$ biologically independent experiments). **k–m** Time course of P5C efflux (**k**), glutamate accumulation (**l**), and glutamate efflux (**m**) in WT and *put6Δ* mitochondria following 1 mM proline addition. All the data points were normalized to time zero and plotted as mean ± SD ($n = 3$ biologically independent experiments). WT and mutants used in this panel are from prototrophic ACY genetic background (see Supplementary Table 1). Source data are provided as a Source data file.

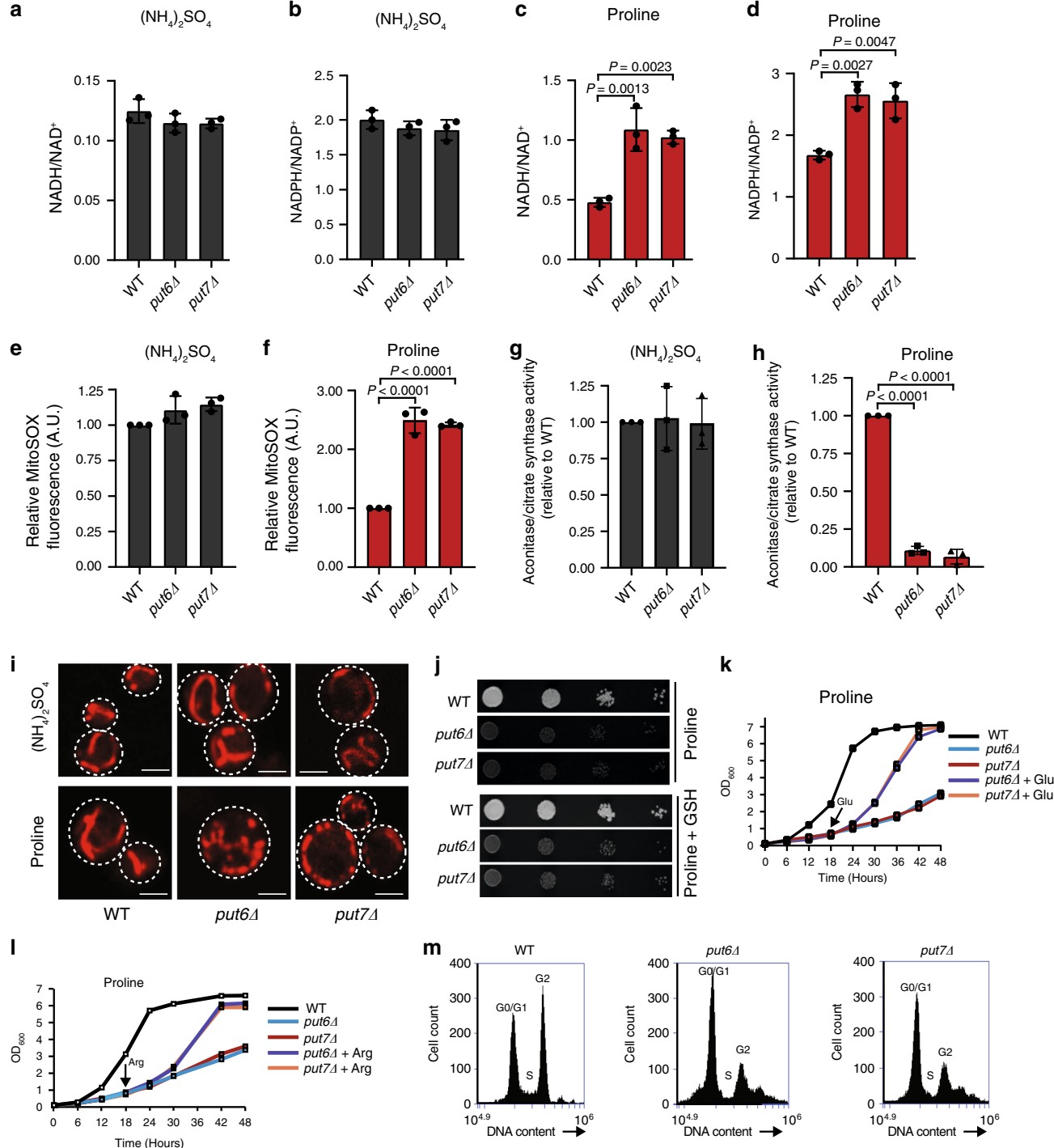

**Fig. 6 Loss of Put6 and Put7 perturbs cellular redox homeostasis. a–d** $NADH/NAD^+$ and $NADPH/NADP^+$ in WT, $put6\Delta$, and $put7\Delta$ yeast cells when grown on synthetic galactose media with ammonium sulfate (**a** and **b**) or proline (**c** and **d**) as the sole nitrogen source. Data are expressed as mean ± SD ($n = 3$ biologically independent experiments). **e**, **f** Mitochondrial ROS measured using MitoSOX fluorescence. Data are expressed as mean ± SD ($n = 3$ biologically independent experiments). **g**, **h** Aconitase activity in isolated mitochondria from WT, $put6\Delta$, and $put7\Delta$ yeast cells. Aconitase activity was normalized to citrate synthase activity and is expressed as mean ± SD relative to WT ($n = 3$ biologically independent experiments). **i** Indicated yeast cells were stained with the Mitotracker Red CMXRos and visualized using confocal microscopy (scale bar 2.44 μm). **j** Ten-fold serially diluted yeast cells were seeded on synthetic media plates containing proline as the sole nitrogen source ± 0.5 mM reduced glutathione (GSH). Pictures were taken after 3 days of seeding. The data are representative of three independent experiments. **k**, **l** Growth measurement of WT, $put6\Delta$, and $put7\Delta$ yeast cells in proline-containing media followed by 4 mM glutamate (**k**) or 4 mM arginine (**l**) addition after 18 h of growth (indicated by an arrow). Data are representative of three independent experiments. **m** Cell-cycle analysis of proline grown WT, $put6\Delta$, and $put7\Delta$ cells by flow cytometry. Statistical significance was assessed by one-way analysis of variance (ANOVA) with Tukey's multiple comparison test. WT and mutants used in this panel are from prototrophic ACY genetic background (see Supplementary Table 1). Source data are provided as a Source data file.

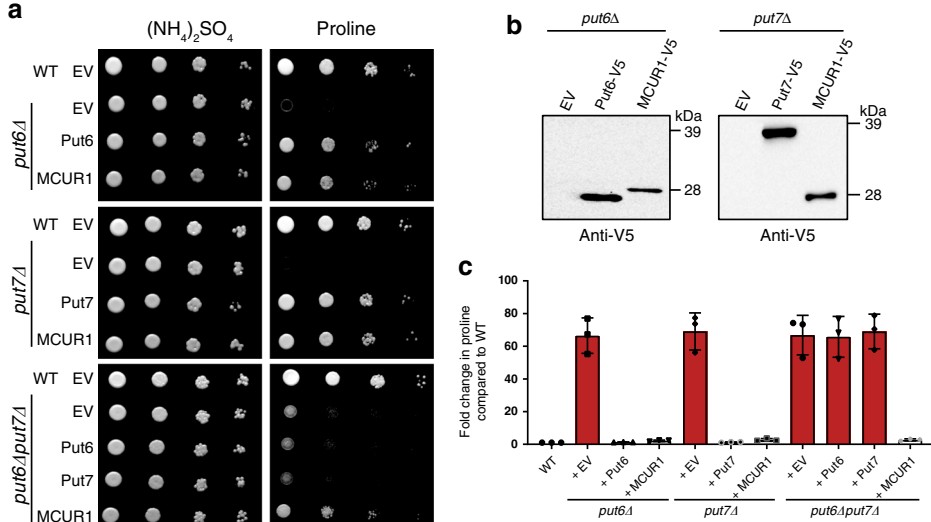

**Fig. 7 MCUR1 can functionally complement *put6Δ* and *put7Δ* yeast mutants. a** Indicated yeast strains were transformed with empty vector, Put6-V5, Put7-V5, or MCUR1-V5 expressing plasmids, serially diluted, and seeded on plates containing ammonium sulfate or proline as the sole nitrogen source. Images were captured after 3 days of growth at 30 °C. The data are representative of three independent experiments. **b** Mitochondria isolated from the indicated cells were subjected to SDS-PAGE immunoblotting analysis using anti-V5 antibody. Data are representative of two independent experiments. **c** Proline levels in WT and indicated yeast mutants cultured in proline-containing media. Relative proline levels were normalized to WT control and plotted as mean ± SD (*n* = 3 biologically independent experiments). WT and mutants used in this panel are from prototrophic CEN.PK genetic background (see Supplementary Table 1). Source data are provided as a Source data file.

for their ability to grow on different non-fermentable carbon sources at varying temperatures. However, we still did not observe any respiratory growth defect, further suggesting that Put6 is not involved in MRC function (Supplementary Fig. 3).

The presence of Put6 and Put7 in the IMM in the form of a hetero-oligomeric complex suggested their possible role in the transport processes across the IMM. Since the identity of a number of predicted IMM amino acid transporters is still unknown[32], we tested for the potential role of Put6 and Put7 in amino acid utilization. In most eukaryotes, proline catabolism exclusively occurs in the mitochondrial matrix; however, the mitochondrial proline import machinery has not been identified[2,33]. Our metabolomics data showing proline accumulation in *put6Δ* cells suggested a potential defect in either mitochondrial proline uptake or its catabolism. Our data are consistent with a previous genome-wide study that reported reduced fitness of *put6Δ* cells that were cultured in medium containing proline as the sole nitrogen source[34]. Therefore, we performed radiolabeled proline uptake assays in isolated mitochondria, which demonstrated that proline uptake is intact in *put6Δ* cells; in fact, proline uptake occurred with faster kinetics in the mutant (Fig. 5j). In addition to the proline import machinery, protein(s) involved in the export of pyrroline-5-carboxylate (P5C), a downstream metabolite of proline catabolism, are yet to be identified[35]. The import of proline and export of P5C from mitochondria constitute the "proline-P5C cycle", which has been proposed to maintain the redox balance between the mitochondria and cytosol[36,37]. Therefore, we considered the possibility that Put6/Put7 are involved in P5C efflux; however, P5C efflux data ruled out this possibility (Fig. 5k). Inability to grow on proline could be due to a downstream defect in distributing nitrogen from proline to other nitrogen requiring pathways. Therefore, we explored the possibility that glutamate efflux from mitochondria is perturbed in cells lacking Put6. Again, we did not observe any defect in glutamate export (Fig. 5m), but we did observe rapid glutamate accumulation in mitochondria (Fig. 5l), which is consistent with rapid proline uptake in the mitochondria. Increased mitochondrial proline uptake in *put6Δ* suggests that Put6/Put7

complex plays a regulatory role in controlling proline catabolism. Controlled utilization of proline is necessary to prevent redox imbalance and generation of ROS, which we observed in *put6Δ* cells.

These findings raise two important questions—what is the precise role of Put6/Put7 in mitochondrial proline utilization and is the role of Put6/Put7 in proline utilization conserved in higher eukaryotes? Since proline is the most abundant nitrogen source in grapes, the natural habitat for yeast cells[38], we speculate that Put6/Put7 play an essential role for nitrogen assimilation in yeast in their natural environment. Proline is unlikely to be the main nitrogen source for most eukaryotes, so these proteins may have evolved to perform other functions in other species. However, our finding showing the rescue of *put6Δ* and *put7Δ* by MCUR1 expression (Fig. 7) strongly suggests that these proteins have a conserved function in mitochondrial proline metabolism and warrants future investigations on the role of MCUR1 in mammalian proline metabolism.

## Methods

**Yeast strains and growth conditions.** *Saccharomyces cerevisiae* strains, plasmids, antibodies, and primers used in this study are listed in the Supplementary Tables 1, 2, 3, and 4 respectively. Single- and double-knockout yeast strains were constructed by one-step gene disruption method using geneticin, hygromycin, or nourseothricin cassettes[39]. Yeast transformation was performed by the standard lithium acetate method, and cells were selected on YPD (1% yeast extract, 2% peptone, and 2% glucose) plates containing appropriate antibiotics or synthetic dropout media. Growth assays were carried out in liquid media or on agar plates following 10-fold serial dilutions at 30 or 37 °C. Growth was monitored spectrophotometrically at 600 nm in a liquid medium or by taking pictures of plates at the indicated time points. Growth measurement on different carbon sources was performed using 2% dextrose (YPD), 2% glycerol (YPG), 2% ethanol (YPE), 3% glycerol, and 1% ethanol (YPGE)-containing media. In addition to the carbon source, the media contained 1% yeast extract and 2% bactopeptone. Growth on different nitrogen sources was performed in synthetic minimal media containing 0.17% yeast nitrogen base (YNB, without amino acids or ammonium sulfate), 2% galactose, and 4 mM of the indicated amino acids or 0.5% ammonium sulfate.

**Phylogenetic analysis.** MCUR1 homologs from 15 selected species were identified by DELTA-BLAST analysis[40]. Multiple sequence analysis of MCUR1 homologs was performed using CLUSTAL omega[41] and phylogenetic tree was generated

using default parameters. The presence of MCU, MICU1, and EMRE in the selected 15 species was determined by BLASTP analysis.

**Mitochondria isolation**. Yeast cells were grown to mid-log phase and harvested by centrifugation. Cells were disrupted by zymolyase treatment at 30 °C, followed by 15 strokes of Dounce homogenization at 4 °C. Following a $4000 \times g$ clarifying spin, mitochondria were collected by centrifugation at $12,000 \times g$[42]. The mitochondrial pellet was resuspended in SEM buffer (250 mM sucrose, 1 mM EDTA, and 10 mM MOPS–KOH, pH 7.2) containing protease inhibitor cocktail (Roche Diagnostic). Mitochondrial protein concentration was determined by the BCA assay (Pierce[TM] BCA Protein Assay, Catalog number: 23225).

**SDS/BN-PAGE and immunoblotting**. Mitochondria solubilized in NuPAGE buffer were resolved by 10 or 12% NuPAGE Bis-Tris gel (Life Technologies) and protein expression and abundance was assessed by immunoblotting. For BN-PAGE[43], yeast mitochondria were solubilized for 15 min on ice in buffer containing 1xNative PAGE buffer and 1% digitonin (Life Technologies). After a clarifying spin at $20,000 \times g$ for 30 min, the cleared lysate was resolved on a 3–12% gradient Native Bis-Tris gel (Life Technologies) and transferred to a PVDF membrane using a Mini-PROTEAN Tetra cell (Bio-Rad) for immunoblotting.

**In-gel activity assays**. Mitochondrial respiratory chain complex activities were determined by clear native (CN) PAGE In-gel activity assays[44]. Yeast mitochondria (30 µg of protein for complex II, 90 µg for complexes III and IV, and 20 µg for complex V) were solubilized for 15 min on ice in solubilization buffer (1xNative PAGE buffer and 1% digitonin). The solubilized fraction was resolved using 4–16% gradient native PAGE gel (Life Technologies) with the following additions to the cathode buffer: for complex II, 0.05% sodium deoxycholate (DOC) and 0.01% n-dodecyl β-d-maltoside (DDM); for complex III, no addition; for complex IV, 0.05% DOC and 0.05% DDM; and for complex V, 0.05% DOC and 0.01% DDM were added. The gels were incubated in their respective activity staining solutions. Equal loading was determined by Coomassie blue staining.

**Subcellular fractionation**. Subcellular localization of mitochondrial protein was performed by differential centrifugation[45]. Yeast cells were harvested and disrupted by zymolyase treatment at 30 °C, followed by 15 strokes of Dounce homogenization at 4 °C. Following a $4000 \times g$ clarifying spin, cellular fractions were collected by sequential differential centrifugation. Cleared lysate from $4,000 \times g$ (total) was spun at $13,000 \times g$ to pellet mitochondria (P13). The remaining supernatant (S13) was then centrifuged at $100,000 \times g$ to separate microsomes (P100) and the cytosolic fraction (S100). Equal proportions of the total, pellet (P13, P100) and supernatant (S13, S100) fractions were analyzed by SDS-PAGE and immunoblotting.

**Carbonate extraction assay**. 100 µg of crude yeast mitochondria were resuspended in 200 µl of 100 mM sodium carbonate, pH 11.5 and incubated on ice for 30 min. The membrane-associated fraction was pelleted by ultracentrifugation at $135,000 \times g$ for 30 min. The pellet was resuspended in 25 µl of 2X SDS sample buffer. The supernatant was transferred to a clean microcentrifuge tube, mixed with 1/5th volume of 72% trichloroacetic acid (TCA) and incubated on ice for 30 min. Precipitated proteins were washed with cold acetone and analyzed by SDS-PAGE and immunoblotting.

**Protease-protection assay**. To determine the sub-mitochondrial localization of proteins, 100 µg of isolated mitochondria were incubated in either isotonic SEM buffer (250 mM sucrose, 1 mM EDTA in 10 mM MOPS–KOH buffer, pH 7.2), swelling buffer (10 mM MOPS–KOH buffer, pH 7.2), or solubilization buffer (0.5% Triton X-100) for 15 min. The samples were then treated with or without proteinase K (50 µg/ml) for 15 min, and the reaction was stopped using 2 mM PMSF. Mitochondrial proteins were then precipitated using TCA and analyzed by SDS-PAGE and immunoblotting.

**Oxygen consumption rate measurement**. Oxygen consumption on YPGE grown cells was measured using high-resolution O2K Fluorespirometer (Oroboros) at 30 °C. For each assay 2.1 ml of YPGE media was added into the O2K chamber and $10^6$ cells were injected. After basal respiration was measured, 2 µM antimycin A was added to measure the mitochondrial respiration. Maximal respiration was determined by adding 5 µM CCCP to the cells.

**Measurement of mitochondrial membrane potential**. For the measurement of mitochondrial membrane potential ($\Delta\psi_m$), cells were stained with tetramethylrhodamine methyl ester (TMRM, Thermo Fisher Scientific), which accumulates in mitochondria due to their transmembrane potential. Approximately $1 \times 10^7$ cells were incubated with 0.2 µM TMRM for 30 min at 30 °C while shaking. Cells were washed and analyzed using BD Accuri[TM] C6 flow cytometer after treatment with or without 10 µM CCCP for 10 min.

**Co-immunoprecipitation assay**. 1 mg of crude mitochondria isolated from yeast cells were solubilized in immunoprecipitation (IP) buffer [20 mM HEPES-KOH, pH 7.4, 1 mM CaCl₂, 100 mM NaCl, 1% digitonin, and 10% glycerol, containing protease inhibitor cocktail (Roche Diagnostics)] for 30 min in a rotator at 4 °C. The insoluble mitochondrial fraction was pelleted at $20,000 \times g$ for 30 min, and the soluble supernatant was used for IP. IP was performed using the Dynabeads® Protein G Immunoprecipitation Kit (Thermo Fisher Scientific; Cat. No. 10007D) as per the manufacturer's protocol. The soluble protein fractions were analyzed by SDS-PAGE/immunoblotting.

**Proteomic analysis of Put6/Put7 interacting proteins**. Mitochondria were isolated from cells grown in synthetic media containing galactose as carbon source and ammonium sulfate as nitrogen source. Three milligram of crude mitochondria isolated from *put6Δ/put7Δ* yeast cells transformed with empty vector or Put6-V5/Put7-V5 were solubilized in IP buffer and immunoprecipitated using anti-V5 antibody chemically cross-linked to Dynabeads® Protein G. Cross-linking was performed by BS³ reagent as per the manufacturer's instructions (Thermo Scientific, USA). Proteins were eluted from the beads with 100 µl 0.1 M glycine, pH 3.0, on a shaker at 25 °C for 5 min. The elution was repeated three times, and the eluates of each sample were pooled. Proteins were precipitated with cold acetone, centrifuged at $16,000 \times g$, air-dried, and resuspended in 50 µl of LYSE buffer (PreOmics). Samples were digested overnight at 37 °C in a shaker after adding 0.5 µg of Trypsin (Sigma-Aldrich) and 0.5 µg Lys-C (Wako, Japan). The digestion was stopped after ~12 h by acidifying the sample with 1% trifluoroacetic acid (TFA) and a final peptide cleanup was performed[46].

Peptides were separated on 50 cm columns of ReproSil-Pur C18-AQ 1.9 µm resin packed in-house. The columns were kept at 60 °C using a column oven controlled by the SprayQC software[47]. Liquid chromatography was performed on an EASY-nLC 1200 ultra-high pressure system coupled through a nanoelectrospray source to a Q Exactive HF-X orbitrap mass spectrometer (Thermo Fisher Scientific). Peptides were separated on a nonlinear 120-min gradient of 2–95% buffer A (0.1% formic acid)-buffer B (0.1% formic acid, 80% acetonitrile) at a flow rate of 250 nl/min. Data acquisition switched between a full scan and 15 data-dependent MS/MS scans. Multiple sequencing of peptides was minimized by excluding the selected peptide candidates for 30 s.

The MaxQuant software (version 1.6.1.13) was used for the analysis of raw files and peak lists were searched against the yeast Uniprot FASTA reference proteomes version of 2014 and a common contaminants database by the Andromeda search engine. False discovery rate was set to 1% for peptides (minimum length of 7 amino acids) and proteins and was determined by searching a reverse database. A maximum of two missed cleavages were allowed in the database search. Peptide identification was performed with an allowed initial precursor mass deviation up to 7 ppm and an allowed fragment mass deviation of 20 ppm. The mass spectrometry proteomics data have been deposited to the ProteomeXchange Consortium via the PRIDE[48] partner repository with the dataset identifier PXD015476.

**Yeast cell-cycle analysis**. Yeast cellular DNA content was measured by staining cells with SYBR green dye followed by flow cytometry[49]. Approximately $1 \times 10^7$ cells were harvested and fixed overnight in 1 ml of 95% ethanol at 4 °C. Fixed cells were harvested by centrifugation at $10,000 \times g$ for 20 min and washed in 0.5 ml of sterile H₂O. Water was removed and cells were resuspended in 200 µl of RNase A solution (10 mg/ml RNase A in 50 mM Tris-Cl, pH 8.0) followed by incubation at 37 °C for 2–4 h. The cells were then collected by centrifugation and resuspended in 200 µl of proteinase K solution (2 mg/ml proteinase K in 50 mM Tris-Cl buffer, pH 7.5) and incubated at 50 °C for 60 min. Cells were then centrifuged and the pellet was washed in 500 µl FACS buffer (200 mM Tris-Cl pH 7.5, 200 mM NaCl, 78 mM MgCl₂). For cell-cycle analysis, the cell pellet was resuspended in 1 ml SYBR green solution (1 µl SYBR Green dye in 1 ml FACS buffer), sonicated at low power and analyzed immediately using a BD Accuri[TM] C6 flow cytometer.

**RNA isolation and quantitative PCR**. RNA was extracted from ~$4.5 \times 10^7$ yeast cells by glass bead beating using the RNeasy mini kit (QIAGEN). Five hundred nanograms of total RNA was used as starting material for cDNA synthesis using SuperScript[TM] III First-Strand Synthesis System Kit (Thermo Fisher Scientific; Cat. No. 18080051). Quantitative real-time PCR was performed on CFX96[TM] Real-Time PCR (Bio-Rad) in a 96-well plate. Twenty microliters of PCR reactions were prepared with 2X mastermix and 20X Taqman assay (*PUT6* assay ID: Sc04120572_s1, *PUT7* assay ID: Sc04148698_s1, *PUT1* assay ID: Sc04147047_s1, and *ACT1* assay ID: Sc04120488_s1) from Applied Biosystems. The mRNA levels were normalized to *ACT1* expression levels.

**Yeast metabolomics sample preparation and MS analysis**. Cells grown in minimal media containing ammonium sulfate, proline, or glutamate as the sole nitrogen source for 18 h were spun down and washed twice with sterile nanopure H₂O. Yeast cell pellets were resuspended at a concentration of 1e5 cells/µL (about 300 µL) in extraction solvent consisting of 8:2 (v/v) methanol:water containing 50 µM $^{13}C_5$$^{15}N_2$-labeled-glutamine internal standard and transferred to 2 mL impact resistant homogenization tubes containing 300 mg of 1 mm zirconium beads. Samples were homogenized using a Precellys 24 tissue homogenizer in three

15-s cycles at 6400 Hz followed by sonication for 2 min. Samples were vortexed at 2000 rpm at 4 °C for 30 s and then allowed to rest at −80 °C for 30 min to allow for protein precipitation. Samples were vortexed again at 2000 rpm at 4 °C for 30 s and then centrifuged at 14,000 rpm at 4 °C for 10 min. Supernatants were transferred into amber HPLC vials with 200 μL glass inserts and stored at 4 °C until analysis. For LC-MS/MS analysis, 1 μL (1e5 cells worth of material) of sample was injected and analyzed[50].

**Measurement of intracellular proline levels**. Proline was determined spectrophotometrically using the ninhydrin assay[51]. Yeast cells grown in indicated media for 18 h were harvested and washed twice with 0.9% NaCl and resuspended in 0.5 ml of sterile nanopure $H_2O$. The cells were transferred to a boiling water bath, and the intracellular amino acids were extracted by boiling for 10 min. The samples were centrifuged at $20,000 \times g$ for 10 min at 4 °C, and the proline levels were determined in the supernatant using ninhydrin reagent. Two hundred microliters of clear supernatant was incubated with 200 μl glacial acetic acid and 200 μl of acid-ninhydrin reagent (250 mg ninhydrin dissolved in 4 ml 6 M phosphoric acid plus 6 ml glacial acetic acid) for 60 min in a boiling water bath. The reaction was stopped by 5-min incubation on ice and the mixture was extracted by vigorous vortexing in 400 μl toluene. Proline-derived chromophore in the toluene phase was aspirated and proline levels were determined by measuring the absorbance at 520 nm.

**Mitochondrial [14C] proline uptake assay**. For mitochondrial radioactive proline uptake assay, cells were cultured in synthetic media, pH 5.5 containing 0.17% yeast nitrogen base, 2% galactose, and 5 mM proline. Mitochondria were isolated as described previously, and resuspended at 5 mg/ml in uptake buffer (0.2 M sucrose, 1 mM $MgCl_2$, 10 mM KCl, 20 mM HEPES-Tris, pH 7.2) and 50 μl aliquots were prepared in 1.5 ml centrifuge tubes. Samples were kept on ice and used within 2 h of mitochondrial isolation.

[14C] Proline uptake was measured at room temperature using the stop inhibitor method[52] with minor modifications. Mitochondria were treated with rotenone and antimycin for 2 min and uptake was initiated by adding 200 μl of import buffer containing [14C] proline. The reaction was terminated at the indicated time points by rapidly adding 500 μl of ice-cold termination buffer (import buffer containing inhibitors of proline transport, 0.05 mM bathophenanthroline plus 0.1 mM mersalyl acid) and mitochondria were spun down at $16,000 \times g$ at 4 °C, washed with 500 μl termination buffer and radioactivity was quantified in the pellet by liquid scintillation counting. Mitochondrial samples treated with inhibitors of proline transport before starting uptake were used as a control to determine inhibitor sensitive uptake, and radioactivity in the control sample was subtracted from all the experimental data points.

**Measurement of mitochondrial glutamate and P5C efflux**. Fifty microliters of freshly isolated mitochondria were treated with rotenone and antimycin for 2 min and uptake was initiated by adding 150 μl cold proline (1 mM proline in uptake buffer). The reaction was terminated at the indicated time points by rapidly adding 500 μl of ice-cold termination buffer and mitochondria were spun down at $16,000 \times g$ at 4 °C. Mitochondrial samples treated with inhibitors before starting uptake was used as a control. The supernatants were transferred to a fresh microcentrifuge tube, and pellets were washed with 500 μl of ice-cold termination buffer. The mitochondrial pellets were suspended in 300 μl nanopure water, and amino acids were extracted by boiling in a water bath for 10 min. The samples were centrifuged at $20,000 \times g$ for 10 min at 4 °C and supernatant was collected for analysis.

Glutamate levels were determined using the Glutamate-Glo™ assay kit (Promega; Cat. No. J7021) as per manufacturer's protocol. Fifty microliters of the mitochondrial supernatant and pellet, obtained as described above, were mixed with 50 μl of glutamate detection reagent in a 96-well plate and incubated for 60 min at room temperature. Luminescence was recorded using a Synergy™ Mx (BioTek) microplate reader. A negative control (buffer only) was used to determine the assay background, and the values were subtracted from the test samples.

P5C levels were determined by monitoring the amount of P5C-ortho-amino benzaldehyde complex at 440 nm[53]. One hundred and fifty microliters of buffer (control) or sample was mixed with 500 μl of o-amino benzaldehyde solution (10 mg/ml in 95% ethanol) and incubated at room temperature for 30 min. The reaction was centrifuged at $15,000 \times g$ for 5 min, and the clear yellow supernatant was used to measure absorbance at 440 nm using the control sample as a blank.

**Determination of NAD/NADH and NADP/NADPH ratio**. Cells grown in ammonium sulfate- or proline-containing minimal synthetic media for 16–18 h were harvested and washed with nanopure water. Spheroplasts were obtained from $10^7$ cells by resuspending cells in 1 ml zymolyase buffer (1.2 M sorbitol and 20 mM potassium phosphate, pH 7.4) containing 0.3 mg/ml zymolyase and incubating for 45 min at 30 °C. Spheroplasts were gently washed using zymolyase buffer and resuspended in 200 μl PBS buffer. NAD, NADH, NADP, and NADPH levels were determined using the NAD/NADH-Glo™ assay and NADP/NADPH-Glo™ assay kits (Promega; Cat. No. G9071 and G9081) according to the manufacturer's

protocol. Luminescence was recorded using a Synergy™ Mx (BioTek) microplate reader. Values were normalized to buffer control.

**Measurement of mitochondrial ROS in yeast cells**. To measure mitochondrial superoxide production, $1 \times 10^7$ yeast cells were loaded with 5 μM MitoSOX Red (Thermo Fisher Scientific) for 30 min at 30 °C, washed and resuspended in PBS. The superoxide levels in cells were then measured using a BD Accuri™ C6 flow cytometer.

**Aconitase/citrate synthase activity assay**. Ten micrograms of isolated mitochondria were resuspended in 170 μl of 100 mM triethanolamine buffer, pH 8.0, containing 1.2 mM $NADP^+$ and 10 μL isocitrate dehydrogenase (Sigma-Aldrich, I2002) in a clear bottom 96-well plate (Falcon). Twenty microliters of cis-aconitate was added (final concentration 240 μM) and reduction of $NADP^+$ was monitored at 340 nm for a total of 10 min. The rate of reaction was calculated from the slope of the linear part of the kinetic curve. As a control, activity was also measured in pure buffer or mitochondrial samples before and after the addition of cis-aconitate[54]. Aconitase activity was normalized to citrate synthase activity and displayed as a percentage of WT activity.

For citrate synthase activity, 20 μg of isolated mitochondria were resuspended in 100 μl of 10 mM Tris-HCl buffer, pH 7.5, 0.2% Triton X-100 (v/v) and 200 μM of 5,5-dithiobis-(2-nitrobenzoic acid) (DTNB), in a clear bottom 96-well plate (Falcon). Fifty microliters of acetyl-CoA (2 mM) was added to this solution. After 5 min of incubation, reaction was started by adding 50 μl of oxaloacetate (2 mM) and turn-over of acetyl-CoA was monitored by observing the absorbance at 412 nm for 10 min. The initial rate of reaction was calculated from the slope of the linear part of the kinetic curve[55].

**Fluorescence microscopy**. Cells grown in minimal media containing ammonium sulfate, or proline as the sole nitrogen source for 18 h were spun down and loaded with 200 nM MitoTracker™ Red CMXRos (Thermo Scientific) dye in the respective media for 45 min at 30 °C shaker in dark. Cells were washed thrice with the media and then mounted on glass slides coated with concanavalin A and imaged immediately using Olympus® FV1000 confocal microscope equipped with an UPLSAPO 100×/1.4 oil immersion objective with confocal pinhole size corresponding to 1 Airy unit.

**Statistical analysis and software**. Statistical analysis was conducted using one-way analysis of variance (ANOVA) with Tukey's multiple comparison test using GraphPad Prism. GraphPad Prism 8 was also used to generate growth curves and graphs. Each replicate is defined as an independent experiment performed on a different day using a different clone. ImageJ was used to quantitatively analyze Western blots and process confocal images. Final figures were prepared using Adobe Illustrator.

**Reporting summary**. Further information on research design is available in the Nature Research Reporting Summary linked to this article.

## Data availability

The mass spectrometry proteomics data are available via ProteomeXchange with identifier PXD015476. Source data are provided with this paper.

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

## Acknowledgements

We thank Dr. Jan Brix, for generously providing us with antibodies. We are grateful to Dr. Matthias Mann for his support in mass spectrometry data generation and analysis. We thank Microscopy and Imaging Center, Texas A&M University, for their assistance with confocal microscopy. We are grateful to Dr. Richard Gomer lab at Texas A&M University for helping with the flow cytometry data generation and analysis. We thank Dr. Craig Kaplan for providing yeast expression vectors. This work was supported by the Welch Foundation Grant (A-1810) and the National Institutes of Health awards R01GM111672 to V.M.G.; S10OD020025 and R01ES027595 to M.J.; K01DK116917 to J.D.W. Y.C. was supported by a UC San Diego Chancellor's Research Excellence Scholarship and R.N. was supported by the Foundation for Strategic Research grant no. FFL12-0220. The content is solely the responsibility of the authors and does not necessarily represent the official views of the National Institutes of Health.

## Author contributions

V.M.G. conceptualized the project and M.Z. and V.M.G. designed the experiments, M.Z., J.K.N., S.A.T., N.M.G., Y.C., J.D.W., M.M., S.K.A., and D.T. performed the experiments, M.Z., J.K.N., N.M.G., P.P.T., J.D.W., R.N., M.M., M.J., and V.M.G. analyzed the data. M.Z. and V.M.G. wrote the manuscript. V.M.G. supervised the whole project and was responsible for the resources and funding acquisition.

## Competing interests

The authors declare no competing interests.
