## [Peer Review File · Nature Communications]

Reviewers' comments:

Reviewer #1 (Remarks to the Author):

In the manuscript entitled "Put6 and Put7 are essential for mitochondrial proline metabolism", the authors convincingly show that the yeast orthologues of the human calcium carrier regulator (MCUR1) are required for proline metabolism. This is particularly interesting since *S. cerevisiae* does not possess a mitochondrial calcium uniporter but has retained MCUR1 orthologues. Moreover, the work presented clarifies an existing discrepancy relative to the role of Put6 and Put7 in OXPHOS biogenesis. The study is well designed, includes a thorough phenotypic analysis and provide novel information on the role of yeast Put6 and Put7 in proline utilization. In my opinion, the manuscript is suitable for publication in Nature Communications. I do however have concerns regarding the lack of insights into the molecular mechanism governing Put6 and Put7 function that should be addressed. Human MCUR1 represses calcium uptake by binding to the calcium uniporter MCU and regulating its function. In the case of Put6 and Put7, the authors did not identify any potential interacting partners by immunoprecipitation and MS analysis. However, it is not specified under which conditions were grown the cells used for mitochondria isolation. Growth in presence or absence of proline could be key to identify interactors which binding to Put6 and Put7 depends on proline metabolism. Indeed, the authors observed an increase in Put6 and Put7 protein steady-state levels and oligomer accumulation in cells grown in proline containing media. Interestingly enough, the increase in Put7 amounts appears to be more modest compare to the 20-fold increase in Put6 protein. The authors also showed that Put6 and Put7 stability depend on each other. This open the possibility that the stoichiometry of Put6 and Put7 in the oligomers changes in response to proline. Moreover, MCUR1 binds divalent cations and the structure of its paralog CCDC90B has been recently resolved. Are the residues involved in cations/calcium binding conserved amongst MCUR1 and Put6 and Put7? Based on sequence homology and/or structure modelling, is it possible to speculate about the possibility of Put6 and Put7 binding cations or small molecules? The IP buffer used contains 1 mM CaCl₂. Could this have an effect of Put6 and Put7 network of interactions? Lastly, weak regulatory interactions could be lost during the immunoprecipitation process. Have the authors considered cross-linking or proximity-dependent labelling approaches?

Minor points:

- Are Puf6 and Puf7 tagged proteins fully functional?
- Does overexpression of PUF6 or PUF7 affects mitochondrial redox state, in particular in conditions characterized by an increase in mitochondrial ROS level?
- Is the interaction of human MCUR1 with Puf6 or Puf7 proteins required for functional complementation of the single puf6 or puf7 mutants? Or is MCUR1 expression able to suppress the defect of a double puf6puf7 knockout?

Reviewer #2 (Remarks to the Author):

MCUR1 has been implicated in MCU-dependent mitochondrial Ca²⁺ homeostasis and MRC complex IV biogenesis in a variety of eukaryotes. Nevertheless, why many fungi lacking MCU still contain MCUR1 homologs still remained unknown. In the current study, the authors identified two previously uncharacterised mitochondrial genes, discovered their involvement in the regulation of mitochondrial proline metabolism and consequently, renamed them Put6 and Put7. This discovery, together with the rescue of put6 Δ and put7 Δ by MCUR1 expression, identifies a role for MCUR1 in proline metabolism, beyond calcium-dependent mitochondrial homeostasis, that might be conserved in higher eukaryotes.

Major comments

1. The authors have extensively characterized the phenotype of the put6 Δ and put7 Δ mutants and have shown Put6 and Put7 proteins localize to the inner mitochondrial membrane, where they form a large hetero-oligomeric complex, whose abundance is regulated by proline. Whilst this provides strong evidence to implicate Put6 & Put7 in the regulation of proline metabolism, the authors do not provide enough evidence to support the mechanism linking Put6 & Put7 to ROS generation, or the claim that Put6 and Put7 are necessary to prevent redox imbalance. Further experiments are needed to support this claim.

2. The growth defect upon proline supplementation is clear but would benefit from additional experiments in SM + glucose background (to clearly show that the growth defect upon proline supplementation is independent of MRC function). The authors' claim that the growth defect is completely independent of defects in mitochondrial respiratory function is not fully supported by their experiments. Although they have demonstrated the physical independence of Put6/7 and 3 out of the 4 ETC complexes, this does not rule out functional interdependence. The claim that growth of Δ put6 mutant strain is rescued in SM + galactose with proline as sole nitrogen source by the addition of arginine (Fig. 4b) seems to contrast with the observations of VanderLuis et al (cited by the authors as reference 26), where the growth of an analogous Δ put6 mutant strain in presence of the two amino acids is similarly impaired in comparison to the wild type reference.

Minor comments

The title suggests that put6 and put7 mutants are auxotrophs, but they only have an effect when cells need to use proline as the sole nitrogen source (which is unlikely in physiological conditions) (figure 4a SM+(NH₄)₂SO₄ vs SM+Proline). Authors should re-write the title to reflect the results.

At line 81, the authors state that they performed a phylogenetic analysis across 15 selected species to demonstrate the presence of the MCUR1 protein (results are shown in Fig 1a). It would be appreciated if they justified the choice of these species.

The authors may need to acknowledge that Put6 and Put7 have individual roles, further to their interaction in a complex, as suggested by:

1. whilst it's clear that Put6 and Put7 interact in a hetero-oligomeric complex, as they co-precipitate with strong statistical significance, the absence of common IP proteins suggests that these proteins individually interact with other proteins/complexes.
2. Presence of smaller complexes in fig 4 g for Put7 but not Put6
3. Effect of proline supplementation on gene expression increase expression of Put6 20 fold compared to just 1.5 fold for Put7

In figure 3 and 4, the authors claim that Put6 and Put7 are not required for MRC, but the experiments don't fully disprove this requirement:

Authors should show how cells respond to growth in SM+Glucose+Proline vs SM+Galactose+Proline.

Why did the authors do all supplementation experiments in SM+galactose instead of SM+glucose? (clarification in the text)

How did the authors choose the 10 amino acids used for the supplementation experiments?

(clarification required in the text). If there is no clear specific reason, then, the authors should perform experiments with the remaining proteinogenic aa, to check whether put6 and put7 affect the metabolism of other amino acids too).

Why did the authors choose to supplement with arginine?

Mechanistic link to Redox imbalance :

As mentioned previously, the experiments conducted to demonstrate a link between Put6 & Put7 and redox imbalance do not offer sufficient proof.

Firstly, a two-fold increase in MitoSox is not particularly striking, especially when accompanied by a simultaneous increase in NADPH/NADH levels. Additionally, previous work (<https://onlinelibrary.wiley.com/doi/epdf/10.1002/yea.3154>) has shown that owing to the antioxidant

properties of proline, its intracellular accumulation can help *S.cerevisiae* cells survive ethanol stress. The increased levels of NADPH & NADH would be consistent with previous work suggesting an increased antioxidant capacity under proline accumulation. However, further experiments are needed to reconcile the increase in ROS observed by the authors with the same.

Perhaps, measuring other indicators of the cellular redox state, such as glutathione mediated antioxidant response will provide further clarity on this paradox.

Secondly, although the authors' claim that ROS or NADPH/NADH imbalance is leading to the observed cell cycle arrest, is very plausible, there is no clear mechanism that can be proposed from their work. In order to claim that the cell cycle arrest is caused by redox imbalance, the authors need to probe mechanisms that are known to directly affect cell cycle progression, for eg. the DNA damage response.

Lastly, the fact that redox equivalents have altered concentrations could directly affect mitochondrial respiratory chain activity. This undermines the authors' claim about the independence of Put6 & Put7 and mitochondrial respiration.

Reviewer #3 (Remarks to the Author):

Human mitochondrial calcium uniporter regulator 1 (MCUR1) is involved in the assembly of mitochondrial complexes in the inner membrane (IM) such as the Ca²⁺ uniporter channel complex and cytochrome c oxidase. Here, the authors conduct a comprehensive study of the MCUR1 yeast homolog Fmp32 and the related protein Ylr283. Previous studies of Fmp32 have indicated it is required for proline metabolism and that it has a conserved role in mitochondrial bioenergetics. The novel findings of this study are that deletion of Fmp32 and Ylr283 block the ability of yeast strains to use proline as a nitrogen source and that proline uptake and accumulation are significantly upregulated. Because of the strong link to proline metabolism, the authors propose to rename Fmp32 and Ylr283 as Put6 and Put7, respectively. Biochemical analysis shows that Put6 and Put7 interact with each other in the mitochondrial matrix and have increased protein levels in the presence of proline. Depletion of Put6/Put7 does not impair mitochondrial bioenergetics and the assembly of respiratory complexes. The authors suggest that the Put6 is negative regulator of proline import and metabolism. Strikingly, human MCUR1 is able to rescue the growth phenotypes of the Put6 and Put7 deletion strains in addition to restoring proline metabolic levels. The mechanism by which proline accumulates to high levels in the Put6/Put7 deletion strains is unclear, however, the results convincingly show that both proteins are critical for proline transport. The findings from the study significantly advance the understanding of human MCUR1 and its conserved function in yeast. The manuscript is of broad interest to investigators in mitochondria research and metabolism.

Some questions for the authors to consider.

1. How confident are the authors that HsMCUR1 has two transmembrane domains? In the alignment the other proteins just have one TM. A 2019 structure paper by Alvarez et al., (<https://doi.org/10.1016/j.str.2018.11.004>), which the authors should include and consider, has just one TM at the C-terminal end of HsMCUR1. The authors should consider updating the cartoon alignment in Figure 1b accordingly. This would also better support the ability of HsMCUR1 to rescue the Put6 and Put7 mutants.
2. MitoSOX is used to conclude that the Put6 and Put7 mutant have higher levels of ROS. The authors should complement this with another method such as aconitase activity assays to assess oxidative stress.
3. The band for MCUR1 is 28 kDa instead of 39 kDa for the full-length protein. What post-translational processing are the authors referring to? More explanation is needed to explain the much lower molecular weight band?

4. Could the authors provide more insight into why MCUR1 alone can complement the double knockout put6put7 strain but individually Put6 and Put7 are not able to complement the put6put7 double knockout (Figure 7)?
5. The growth profiles of the Put6 and Put7 deletion strains are comparable to wild-type and there is no disruption in the formation of mitochondrial respiratory complexes. Even so, have the authors confirmed that the mitochondrial membrane potential is similarly unperturbed? TMRM staining of mitochondria in the Put6 and Put7 deletion strains would help confirm this.
6. With the arrest of the put6 and put7 mutants at G1, did the authors examine whether there were any problems with mitochondrial dynamics that may contribute to the cell cycle delay. Are more fragmented mitochondria observed in the mutant strains? These data could provide additional support for the conclusion by the authors on Line 263-265 that "perturbation in the cellular and mitochondrial redox state may explain the growth arrest of the mutants in the proline-containing medium."
7. One line 146 and 147, the authors state "Tomar et al., did not observe any perturbation in any MRC complexes in MCUR1 deficient cells." Previous work by Tomar did show MCUR1 deletion impairs mitochondrial bioenergetics, whether loss of MCUR1 also disrupted mitochondrial respiratory complexes is not clear as it does not seem to have been directly tested. The authors should clarify the point they are trying to make in regards to the paper by Tomar et al.
8. Figure 1, replace "mitochondrial" with "mitochondrial fraction"

Point by Point Response

Reviewer1:

General: In the manuscript entitled "Put6 and Put7 are essential for mitochondrial proline metabolism", the authors convincingly show that the yeast orthologues of the human calcium carrier regulator (MCUR1) are required for proline metabolism. This is particularly interesting since *S. cerevisiae* does not possess a mitochondrial calcium uniporter but has retained MCUR1 orthologues. Moreover, the work presented clarifies an existing discrepancy relative to the role of Put6 and Put7 in OXPHOS biogenesis. The study is well designed, includes a thorough phenotypic analysis and provide novel information on the role of yeast Put6 and Put7 in proline utilization. In my opinion, the manuscript is suitable for publication in Nature Communications.

Response: We thank the reviewer for finding our manuscript "suitable for publication in Nature Communications."

Major Comments:

1. I do however have concerns regarding the lack of insights into the molecular mechanism governing Put6 and Put7 function that should be addressed. Human MCUR1 represses calcium uptake by binding to the calcium uniporter MCU and regulating its function. In the case of Put6 and Put7, the authors did not identify any potential interacting partners by immunoprecipitation and MS analysis. However, it is not specified under which conditions were grown the cells used for mitochondria isolation. Growth in presence or absence of proline could be key to identify interactors which binding to Put6 and Put7 depends on proline metabolism. Weak regulatory interactions could be lost during the immunoprecipitation process. Have the authors considered cross-linking or proximity-dependent labelling approaches?

Response: The reviewer raises an important point. In the original submission, immunoprecipitation (IP) and mass spectrometry (MS) analyses were performed using mitochondria isolated from cells that were grown in synthetic media where ammonium sulfate was used as the sole nitrogen source. We now clearly describe these growth conditions in the Methods Section of the revised manuscript.

Our choice of media was guided by the fact that Put6 and Put7 mutants grow poorly in the media-containing proline as the sole nitrogen source. However, we understand and agree with reviewer's suggestion that proline supplementation may

uncover novel interacting partners of Put6 and Put7. Therefore, we have now performed IP in proline conditioned growth media. Although the cell growth and mitochondrial yield was very low under this condition, we utilized 36 liters of growth media to obtain sufficient mitochondria for the IP/MS experiment. IP/MS analysis of the proline-cultured cells again identified Put6 and Put7 as the two interacting partners of the complex and did not uncover any additional common interacting proteins (See Figure below). These data are consistent with our original findings and further strengthens the existence of a higher order heterologomeric complex of Put6 and Put7 in the mitochondria. The size of the Put6/Put7-containing higher order complex in the ammonium sulfate-grown cells (Fig. 2a) is comparable to that of proline-grown cells (Fig. 4f), which is consistent with the new IP/MS results.

Notably, Coiled-Coil Domain (CCD) containing proteins (like Put6 and Put7) are known to form these types of heterologomeric complexes in mitochondrial membranes as exemplified by a recent cryoEM study showing that MCU, a CCD-containing protein, forms a hetero-octameric complex with EMRE, its interacting partner (Zhou et al. 2020. [bioRxiv. doi.org/10.1101/2020.04.04.025205](https://doi.org/10.1101/2020.04.04.025205)).

2. The authors observed an increase in Put6 and Put7 protein steady-state levels and oligomer accumulation in cells grown in proline containing media. Interestingly enough, the increase in Put7 amounts appears to be more modest compare to the 20-fold increase in Put6 protein. The authors also showed that Put6 and Put7 stability depend on each other. This open the possibility that the stoichiometry of Put6 and Put7 in the oligomers changes in response to proline.

Response: Our data fully supports reviewer's hypothesis. Indeed, Put7 appears to be highly expressed under ammonium sulfate or proline supplementation, whereas Put6 abundance markedly increases in cells grown in proline containing media (Fig. 4d & e). In addition, higher order subcomplexes accumulate only in Put7-V5 and not Put6-V5 expressing cells when grown in proline conditions (Fig. 4f). Based on these results we speculate that Put7 oligomerizes first and then Put6 is recruited at later stages of complex formation. However, with these experimental approaches we cannot determine the stoichiometry of the protein complex and further experiments are needed to define the stoichiometry of Put6 and Put7 containing complexes in different nitrogen sources.

3. Moreover, MCUR1 binds divalent cations and the structure of its paralog CCDC90B has been recently resolved. Are the residues involved in cations/calcium binding conserved amongst MCUR1 and Put6 and Put7? Based on sequence homology and/or structure modelling, is it possible to speculate about the possibility of Put6 and Put7 binding cations or small molecules?

Response: Recently, Adlakha et al, 2019 (PMID: 30612859) showed that MCUR1 can bind divalent cations, in vitro. However, they did not identify Ca²⁺ binding residues. In the absence of this information, we cannot speculate cation/Ca²⁺-binding residues in Put6 or Put7 proteins.

4. The IP buffer used contains 1 mM CaCl₂. Could this have an effect of Put6 and Put7 network of interactions?

Response: Adlakha et al, 2019 (PMID: 30612859) reported "purified truncated MCUR1 forms translucent gel like structure at room temperature over 24 hour time period in a Ca²⁺-dependent manner." However, in our case the IP was carried out at much lower temperature (4°C) for short duration only. Therefore, we believe that Ca²⁺ in buffer is unlikely to have an effect on the interactions under *in vivo* conditions.

Minor points:

1. Are Puf6 and Puf7 tagged proteins fully functional?

Response: We carefully tested the functionality of each of the tagged proteins by performing an *in vivo* complementation assay using *put6Δ* and *put7Δ* cells. As seen in the figure below and in the new supplementary Fig. 4a of the revised manuscript, V5-tagged Put6 and V5- and HA-tagged Put7 proteins used in the study are fully functional as they are able to rescue the growth of *put6Δ* and *put7Δ* cells, respectively, in the proline containing media. Notably, unlike tagged Put7, HA- or myc- tagged Put6 are not functional as they are unable to restore growth of *put6Δ* cells (see figure below and supplementary Fig. 4a of the revised manuscript). Therefore, we only used Put6-V5 in this study.

2. Does overexpression of PUT6 or PUT7 affects mitochondrial redox state, in particular in conditions characterized by an increase in mitochondrial ROS level?

Response: As suggested by reviewer, we have now overexpressed Put6 and Put7 using multicopy plasmids (pRS42N and pRS42H) and found that overexpression of PUT6 and PUT7 leads to a minor growth defect when grown on proline as the sole nitrogen source. This overexpression also led to a significant increase in mitochondrial

ROS in both Put6 and Put7 overexpressing cells that are cultured in proline-containing media (below and in supplementary Fig. 6i & j). These data suggest that optimal levels of Put6 and Put7 are necessary for maintaining mitochondrial ROS homeostasis and are consistent with our proposed role of these proteins in cellular proline metabolism.

3. Is the interaction of human MCUR1 with Put6 or Put7 proteins required for functional complementation of the single *put6* or *put7* mutants? Or is MCUR1 expression able to suppress the defect of a double *put6put7* knockout?

Response: MCUR1 expression alone was able to suppress the growth defect of *put6Δput7Δ* double knockout. See below and new Fig. 7a in the revised manuscript.

Reviewer2:

MCUR1 has been implicated in MCU-dependent mitochondrial Ca^{2+} homeostasis and MRC complex IV biogenesis in a variety of eukaryotes. Nevertheless, why many fungi lacking MCU still contain MCUR1 homologs still remained unknown. In the current study, the authors identified two previously uncharacterized mitochondrial genes, discovered their involvement in the regulation of mitochondrial proline metabolism and consequently, renamed them Put6 and Put7. This discovery, together with the rescue of *put6Δ* and *put7Δ* by MCUR1 expression, identifies a role for MCUR1 in proline metabolism, beyond calcium-dependent mitochondrial homeostasis, that might be conserved in higher eukaryotes.

Response: We thank Reviewer 2 for their thoughtful comments and for recognizing the importance of the work.

Major Comments:

1. The authors have extensively characterized the phenotype of the *put6Δ* and *put7Δ* mutants and have shown Put6 and Put7 proteins localize to the inner mitochondrial membrane, where they form a large hetero-oligomeric complex, whose abundance is regulated by proline. Whilst this provides strong evidence to implicate Put6 & Put7 in the regulation of proline metabolism, the authors do not provide enough evidence to support the mechanism linking Put6 & Put7 to ROS generation, or the claim that Put6 and Put7 are necessary to prevent redox imbalance. Further experiments are needed to support this claim.

Response: Previous studies have shown that increased mitochondrial proline oxidation is accompanied with increased mitochondrial ROS generation (Donald et al., 2001; PMID: 11280728). This is likely due overflow of electrons to the electron transport chain. (Hancock et al., 2016; PMID: 26660760). We recognize that we only used MitoSOX-based readout for ROS level measurements. Therefore, to further support our findings, we have now performed two additional experiments to demonstrate that mitochondrial ROS increases in *put6Δ* and *put7Δ* mutants when grown in media with proline as the sole nitrogen source.

(i) Oxidative inactivation of mitochondrial aconitase: Since mitochondrial aconitase is sensitive to increased ROS levels, we measured aconitase activity in mitochondria isolated from WT, *put6Δ* and *put7Δ* yeast cells grown in ammonium sulfate or proline as the sole nitrogen source. As shown below and in Fig. 6g & h of the revised manuscript, aconitase activity is drastically reduced in *put6Δ* and *put7Δ* yeast cells only when proline is used as the sole nitrogen source.

(ii) Oxidative stress induced mitochondrial fragmentation: Mitochondria undergo fragmentation during oxidative stress (Willems et al., 2015 PMID: 26166745). Therefore, we also checked the mitochondrial morphology of WT and *put6Δ* and *put7Δ* mutants using Mitotracker™ Red after growing cells in ammonium sulfate or proline as the sole

nitrogen source. As shown here and in Fig. 6i of the revised manuscript, the mutants displayed mitochondrial fragmentation when grown in proline.

2. The growth defect upon proline supplementation is clear but would benefit from additional experiments in SM + glucose background (to clearly show that the growth defect upon proline supplementation is independent of MRC function). The authors' claim that the growth defect is completely independent of defects in mitochondrial respiratory function is not fully supported by their experiments.

Response:

We have now performed the suggested experiment and found that growth defects of *put6Δ* and *put7Δ* mutants in proline persist in glucose-containing synthetic medium, suggesting that growth defect in proline is independent of mitochondrial respiratory function (see figure below and Fig. 4b & c in the revised manuscript).

3. Although they have demonstrated the physical independence of Put6/7 and 3 out of the ETC complexes, this does not rule out functional interdependence.

Response:

The reviewer's assertion is correct that physical independence of Put6/7 complex from the ETC complexes does not rule out functional interdependence. We did consider this possibility and had shown that activities of the ETC complexes and ATP synthase are intact in Put6 mutant (Fig. 3e of the original manuscript). To further exclude the involvement of Put6/Put7 complex in ETC function, we measured cellular oxygen

consumption under basal conditions, upon 2 μM antimycin treatment to inhibit mitochondrial respiration, and upon 5 μM CCCP treatment to observe maximal respiration under uncoupled conditions. As seen in the accompanying figure and Fig. 3F of the revised manuscript, our data revealed that mitochondrial, non-mitochondrial, and maximal respiration in *put6 Δ* cells is equivalent to the WT cells. These results clearly show that the loss of Put6 do not impair ETC function.

4. The claim that growth of *put6 Δ* mutant strain is rescue in SM + galactose with proline as sole nitrogen source by the addition of arginine (Fig. 4b) seems contrast with the observations of VanderLuis et al (cited by the authors as reference 26), where the growth of an analogous *put6 Δ* mutant strain in presence of the two amino acids is similarly impaired in comparison to the wild type reference.

Response: The report from VanderSlius et al., 2014 (PMID: 24721214) was a result of high-throughput growth measurements of an entire prototrophic deletion collection. There is high likelihood of artifacts in such high-throughput studies. With that in mind, we carefully performed single gene studies on experimentally validated yeast mutants to show that the growth defect of Put6 and Put7 is observed only in proline-containing media out of all the 20 amino acids tested (Fig. 4a). In our case, we measured growth of

cells in 7 mL media with a starting OD_{600nm} of 0.1 and maximal OD_{600nm} of ~5.0 over 30 hours. Thus, our low-throughput growth assay accounted for ~5-6 doublings. In contrast, VanderSlius et al., performed a 96-well plate based high-throughput growth assay measuring only two doublings from 0.1 to 0.5 over the period of 30-40 hours. Thus, we attribute differences in results from the two studies to differences in the experimental setup. Additionally, we constructed Put6 in different genetic background (CEN.PK) and further show that *put6Δ* mutants display a growth defect in proline that can be rescue by arginine supplementation. Together, our data show that Put6 is specifically required for proline utilization and this phenotype is observed in different prototrophic yeast strains.

Minor comments:

1. The title suggests that Put6 and Put7 mutants are auxotroph, but they only have an effect when cells need to use proline as the sole nitrogen source (which is unlikely in physiological conditions) (figure 4a SM+(NH₄)₂SO₄ vs SM+Proline). Authors should re-write the title to reflect the results.

Response: We have reworded our title to: “**Put6 and Put7 are novel regulators of mitochondrial proline metabolism**”.

2. At line 81, the authors state that they performed a phylogenetic analysis across 15 selected species to demonstrate the presence of the MCUR1 protein (results are shown in Fig 1a). It would be appreciated if they justified the choice of these species.

Response: Our selection of 15 species for phylogenetic analyses was based on two criteria: First, their presence in different phyla across the eukaryotic kingdom and second they are commonly used model organisms for biomedical/genetic research. We have now modified our text as follows “To survey the distribution of MCUR1 across different phyla in eukaryotes, we performed phylogenetic analysis of the MCUR1 protein in 15 commonly used model organisms.”

3. The authors may need to acknowledge that Put6 and Put7 have individual roles, further to their interaction in a complex, as suggested by:

1. whilst it's clear that Put6 and Put7 interact in a hetero-oligomeric complex, as they co-precipitate with strong statistical significance, the absence of common IP proteins suggests that these proteins individually interact with other proteins/complexes.
2. Presence of smaller complexes in fig 4 g for Put7 but not Put6
3. Effect of proline supplementation on gene expression increase expression of Put6 20- fold compared to just 1.5-fold for Put7.

Response: In order to better define the Put6 and Put7 interactome, we have now performed IP-MS experiments from the cells cultured in proline-containing media. The analysis again identified Put6 and Put7 as the two main interacting partners of the complex (See Figure below).

Thus, since Put6 and Put7 are part of a large hetero-oligomeric complex and their loss results in almost identical cellular and biochemical phenotypes, we predict that Put6 and Put7 participate in the similar biological activity. However, the reviewer makes a valid point, based on our data we cannot exclude individual roles of Put6 and Put7 in different cellular processes.

4. In figure 3 and 4, the authors claim that Put6 and Put7 are not required for MRC, but the experiments don't fully disprove this requirement: Authors should show how cells respond to growth in SM+Glucose+Proline vs SM+Galactose+Proline.

Why did the authors do all supplementation experiments in SM+galactose instead of SM+glucose? (clarification in the text)

Response: As mentioned earlier, we have now performed the suggested experiment and show that growth defect of *put6Δ* and *put7Δ* mutants in proline persist in glucose-containing synthetic medium (Fig. 4b & c in the revised manuscript).

To justify the use of different carbon sources we have now added the following sentence in the text: “In these experiments, we chose to use galactose as a carbon source because unlike glucose it promotes respiro-fermentative growth and does not inhibit mitochondrial biogenesis in *S. cerevisiae*.”

5. How did the authors choose the 10 amino acids used for the supplementation experiments? (clarification required in the text). If there is no clear specific reason, then, the authors should perform

experiments with the remaining proteinogenic aa, to check whether *put6* and *put7* affect the metabolism of other amino acids too).

Response: We have now performed the supplementation experiments with the remaining 10 amino acids (see figure below and figure 4a in the revised manuscript) and found that *put6Δ* and *put7Δ* mutants show growth defect only when proline was used as sole nitrogen source. Consistent with the previous study (Ljungdahl et. al., 2012 PMID: 22419079), we found that histidine, lysine, and cysteine cannot be used by yeast as sole nitrogen source (see figure below and figure 4a in the revised manuscript).

Fig. 4 (a) Prototrophic WT, *put6Δ*, and *put7Δ* yeast cells were cultured at 30°C in synthetic liquid media containing the indicated amino acids as the sole nitrogen source. Growth was monitored by measuring absorbance at 600nm at the indicated time intervals. Ammonium sulfate was used as the most favoured nitrogen source.

6. Why did the authors choose to supplement with arginine?

Response: We used glutamate and arginine because they are favorable nitrogen sources and are downstream products of proline oxidation.

7. Mechanistic link to Redox imbalance: As mentioned previously, the experiments conducted to demonstrate a link between Put6 & Put7 and redox imbalance do not offer sufficient proof. Firstly, a two-fold increase in MitoSox is not particularly striking, especially when accompanied by a simultaneous increase in NADPH/NADH levels.

Additionally, previous work

(<https://onlinelibrary.wiley.com/doi/epdf/10.1002/yea.3154>) has shown that owing to the antioxidant properties of proline, its intracellular accumulation can help *S. cerevisiae* cells survive ethanol stress. The increased levels of NADPH & NADH would be consistent with previous work suggesting an increased antioxidant capacity under proline accumulation. However, further experiments are needed to reconcile the increase in ROS observed by the authors with the same.

Perhaps, measuring other indicators of the cellular redox state, such as glutathione mediated antioxidant response will provide further clarity on this paradox.

Secondly, although the authors' claim that ROS or NADPH/NADH imbalance is leading to the observed cell cycle arrest, is very plausible, there is no clear mechanism that can be proposed from their work. In order to claim that the cell cycle arrest is caused by redox imbalance, the authors need to probe mechanisms that are known to directly affect cell cycle progression, for eg. the DNA damage response.

Response: The reviewer raises a critical point. Elevated NAD(P)H levels and proline accumulation in *put6Δ* and *put7Δ* mutants appear paradoxical to the increased mitochondrial ROS levels we observed. Therefore, to validate our findings, we performed additional experiments including aconitase activity assay and show that both *put6Δ* and *put7Δ* mutants indeed have higher mitochondrial ROS when grown in proline as the sole nitrogen source (See our response to this reviewer's major Comment 1). The elevated mitochondrial ROS we observed is consistent with rapid mitochondrial uptake of proline and its oxidation via ETC. Indeed, a previous study has shown that increased mitochondrial proline oxidation by Put1 is accompanied with increased mitochondrial ROS generation (Donald et al., 2001; PMID: 11280728). The study by Takagi et al.,

2016, which is referred by this reviewer, used *put1Δ* cells. The loss of Put1 prevents transfer of electrons from proline to the mitochondrial ETC and thus would not cause an increase in ROS. However, in our experimental set up Put1 is intact, thus with its excess substrate available, an elevation in ROS is expected.

To test whether redox imbalance and increased ROS is the cause of cell cycle arrest, we performed glutathione supplementation experiment. We found that glutathione is not able to rescue the growth defect of *put6Δ* and *put7Δ* cells (Fig. 6j in the revised manuscript and below). We also tested other antioxidants including ascorbic acid and N-

acetyl cysteine, both of which failed to rescue *put6Δ* and *put7Δ* growth defect in proline-containing medium. Together, these results suggest that the oxidative stress is not the primary cause of cell cycle arrest.

However, we can rescue this growth defect upon supplementation with alternate nitrogen source (arginine or glutamate) (Fig. 6k & l in revised manuscript and above), suggesting that the primary cause of cell cycle arrest in *put6* and *put7* mutants is nitrogen starvation likely due to the inability to utilize nitrogen from proline. Our cell cycle data showing that *put6Δ* and *put7Δ* mutants grown in proline are mostly stuck in G0/G1 phase (Fig. 6m and above) is consistent with the previous studies showing that yeast cells grown under nitrogen limiting conditions caused G0/G1 phase arrest (Su et al., 1996; J. Cell Sci.

PMID: 8799823; An et al., 2014, Autophagy, PMID: 25126732). Based on these results, we have now modified the text accordingly.

8. Lastly, the fact that redox equivalents have altered concentrations could directly affect mitochondrial respiratory chain activity. This undermines the authors' claim about the independence of Put6 & Put7 and mitochondrial respiration.

Response: In the revised manuscript, we clearly show that Put6 and Put7 are dispensable for mitochondrial respiratory chain formation or function (Fig. 3). The alteration in redox equivalents are seen only when proline is used as sole nitrogen source. Therefore, Put6/Put7 are not expected to impact mitochondrial respiratory chain function in complex media where variety of nitrogen sources is available, the conditions that were used in Fig. 3. Our data shows that Put6 and Put7 do not impact mitochondrial carbon-based energy metabolism but they impact nitrogen metabolism, specifically pertaining to proline utilization.

Reviewer 3:

Human mitochondrial calcium uniporter regulator 1 (MCUR1) is involved in the assembly of mitochondrial complexes in the inner membrane (IM) such as the Ca²⁺ uniporter channel complex and cytochrome c oxidase. Here, the authors conduct a comprehensive study of the MCUR1 yeast homolog Fmp32 and the related protein Ylr283. Previous studies of Fmp32 have indicated it is required for proline metabolism and that it has a conserved role in mitochondrial bioenergetics. The novel findings of this study are that deletion of Fmp32 and Ylr283 block the ability of yeast strains to use proline as a nitrogen source and that proline uptake and accumulation are significantly upregulated. Because of the strong link to proline metabolism, the authors propose to rename Fmp32 and Ylr283 as Put6 and Put7, respectively. Biochemical analysis shows that Put6 and Put7 interact with each other in the mitochondrial matrix and have increased protein levels in the presence of proline. Depletion of Put6/Put7 does not impair mitochondrial bioenergetics and the assembly of respiratory complexes. The authors suggest that the Put6 is negative regulator of proline import and metabolism. Strikingly, human MCUR1 is able to rescue the growth phenotypes of the Put6 and Put7 deletion strains in addition to restoring proline metabolic levels. The mechanism by which proline accumulates to high levels in the Put6/Put7 deletion strains is unclear, however, the results convincingly show that both proteins are critical for proline transport. The findings from the study significantly advance the understanding of human MCUR1 and its conserved function in yeast. The manuscript is of broad interest to investigators in mitochondria research and metabolism.

Response: We are glad that this Reviewer finds that our “manuscript is of broad interest to investigators in mitochondria research and metabolism” and that this “study significantly advance the understanding of human MCUR1 and its conserved function in yeast.”

Comments:

1. How confident are the authors that HsMCUR1 has two transmembrane domains? In the alignment the other proteins just have one TM. A 2019 structure paper by Alvarez et al., (<https://doi.org/10.1016/j.str.2018.11.004>), which the authors should

include and consider, has just one TM at the C-terminal end of HsMCUR1. The authors should consider updating the cartoon alignment in Figure 1b accordingly. This would also better support the ability of HsMCUR1 to rescue the Put6 and Put7 mutants.

Response: We depicted two transmembrane (TM) domains in MCUR1 based on three sources. First, Mallilankaraman et al., 2012 (PMID: 23178883) used proteinase K treatment to show that MCUR1 has two TM domains. Second, Tomar et al., 2016 (PMID: 27184846) used BiFc complementation assay showing that both N and C terminal of MCUR1 are facing mitochondrial intermembrane space, implying that it has two TM domains. Finally, Uniprot server (<https://www.uniprot.org>) also indicates presence of two transmembrane domains in HsMCUR1. Alvarez et al., used in silico-modeling to predict that CCDC90B homologues including MCUR1 have one TM domain. In our Fig.1b cartoon, we chose to use experimentally validated results for depicting two TM of MCUR1.

2. MitoSOX is used to conclude that the Put6 and Put7 mutant have higher levels of ROS. The authors should complement this with another method such as aconitase activity assays to assess oxidative stress.

Response: As suggested by the reviewer we have now measured aconitase activity in mitochondria isolated from WT, *put6Δ* and *put7Δ* yeast cells grown in ammonium sulfate or proline as sole nitrogen source. As shown here and in Fig. 6g & h in the revised manuscript, aconitase activity was severely reduced in *put6Δ* and *put7Δ* yeast cells when grown in proline containing media. These data complement our MitoSOX data showing elevated mitochondrial ROS levels in these mutants.

3. The band for MCUR1 is 28 kDa instead of 39 kDa for the full-length protein. What post-translational processing are the authors referring to? More explanation is needed to explain the much lower molecular weight band?

Response: In the paper by Alvarez et al, 2019, the authors reported proteolytic processing of MCUR1, where 1-140 amino acid residues were absent in the processed MCUR1. Based on this literature as well as previous study by Tomar et al., 2016 (PMID: 27184846), we have now added the explanation in the revised manuscript that reads as follows “Notably, similar post-translational modification of native MCUR1 has been observed in mammalian cell lines where the processed form lacked the first 140 amino acid residues at N-terminus.”

4. Could the authors provide more insight into why MCUR1 alone can complement the double knockout *put6put7* strain but individually Put6 and Put7 are not able to complement the *put6put7* double knockout (Figure 7)?

Response: As shown in Fig. 2b, loss of Put6 results in loss of Put7 and vice versa. Because expression of each protein is dependent on the presence of other, expressing Put6 or Put7 individually in *put6Δput7Δ* double knockout is not able to rescue proline phenotypes (Fig. 7a & c). However, MCUR1 expression is independent of the presence of either Put6 and Put7 (Fig. 7b), therefore MCUR1 alone is able to complement *put6Δput7Δ* double knockout (Fig, 7a & c).

5. The growth profiles of the Put6 and Put7 deletion strains are comparable to wild-type and there is no disruption in the formation of mitochondrial respiratory complexes. Even so, have the authors confirmed that the mitochondrial membrane potential is similarly unperturbed? TMRM staining of mitochondria in the Put6 and Put7 deletion strains would help confirm this.

Response: As suggested by the reviewer, we have now measured the mitochondrial membrane potential using TMRM fluorescence in WT, *put6Δ* and *put7Δ* yeast cells grown in respiratory growth media and found that there was no significant difference in the TMRM fluorescence in WT and mutants (See figure below and Fig. 3f of the revised manuscript). These results are consistent with our data that levels of MRC complexes or their activities are unperturbed in *put6Δ* and *put7Δ* yeast cells cultured in respiratory growth conditions.

6. With the arrest of the *put6Δ* and *put7Δ* mutants at G1, did the authors examine whether there were any problems with mitochondrial dynamics that may contribute to the cell cycle delay. Are more fragmented mitochondria observed in the mutant strains? These data could provide additional support for the conclusion by the authors on Line 263-265 that “perturbation in the cellular and mitochondrial redox state may explain the growth arrest of the mutants in the proline-containing medium.”

Response: As suggested by the reviewer, we checked the mitochondrial morphology of WT and mutants grown in ammonium sulfate or proline as the sole nitrogen source using Mitotracker™ Red. The mutants displayed mitochondrial fragmentation when grown in proline (below and Fig. 6i in the revised manuscript). This data is consistent with ROS induced perturbation in mitochondrial morphology as predicted by the reviewer.

7. One line 146 and 147, the authors state “Tomar et al., did not observe any perturbation in any MRC complexes in MCUR1 deficient cells.” Previous work by Tomar did show MCUR1 deletion impairs mitochondrial bioenergetics, whether loss of MCUR1 also disrupted mitochondrial respiratory complexes is not clear as it does not seem to have been directly tested. The authors should clarify the point they are trying to make in regards to the paper by Tomar et al.

Response: Indeed, Tomar et al., observed a calcium dependent bioenergetics defect but they did not observe any perturbation in the steady state levels of individual subunits of MRC complexes. Accordingly, to more accurately reflect these data from Tomar et. al, we have modified our text which now reads as “Tomar et al., did not observe any perturbation in the subunits of different MRC complexes in MCUR1 deficient cells.”

8. Figure 1, replace “mitochondrial” with “mitochondrial fraction”

Response: We have now replaced mitochondrial with mitochondrial fraction in Figure 1 legend.

REVIEWERS' COMMENTS:

Reviewer #1 (Remarks to the Author):

In this revised version of the manuscript entitled "Puf6 and Puf7 are novel regulators of mitochondrial proline metabolism", the authors have included a large amount of additional experimental work in support of their conclusions and in my opinion they have addressed the concerns raised by the reviewers.

Reviewer #2 (Remarks to the Author):

The authors have thoroughly revised their paper, conducted many additional experiments and analyses, and satisfactorily answered all my comments. I congratulate them to a nice paper.

Reviewer #3 (Remarks to the Author):

The authors have carefully considered and responded to the critiques from the previous review. A number of new experimental data have been added that strengthen the manuscript and more fully demonstrate the role of Put6 and Put7 in mitochondria. I have no further suggestions for the authors.

Point-by-point response to referees comments:

We are glad that all three referees are satisfied with our responses to their original questions and find our revised manuscript acceptable for publication without any additional change.

Reviewer #1 (Remarks to the Author):

In this revised version of the manuscript entitled "Puf6 and Puf7 are novel regulators of mitochondrial proline metabolism", the authors have included a large amount of additional experimental work in support of their conclusions and in my opinion they have addressed the concerns raised by the reviewers.

Reviewer #2 (Remarks to the Author):

The authors have thoroughly revised their paper, conducted many additional experiments and analyses, and satisfactorily answered all my comments. I congratulate them to a nice paper.

Reviewer #3 (Remarks to the Author):

The authors have carefully considered and responded to the critiques from the previous review. A number of new experimental data have been added that strengthen the manuscript and more fully demonstrate the role of Put6 and Put7 in mitochondria. I have not further suggestions for the authors.